# Multi-Year ACSM measurements at the Central European Research Station Melpitz (Germany) Part I: Instrument Robustness, Quality Assurance, and Impact of Upper Size Cut-Off Diameter

Laurent Poulain[1], Gerald Spindler[1], Achim Grüner[1], Thomas Tuch[1], Bastian Stieger[1], Dominik van Pinxteren[1], Jean-Eudes Petit[2,3], Olivier Favez[4,3], Hartmut Herrmann[1], and Alfred Wiedensohler[1]

[1] Leibniz Institute for Tropospheric Research (TROPOS), Permoserstr. 15, 04317 Leipzig, Germany
[2] Laboratoire des Sciences du Climat et de l'Environnement, CEA-CNRS-UVSQ, IPSL, Université Paris-Saclay, 91191 Gif-sur-Yvette, France
[3] Aerosol Chemical Monitor Calibration Centre (ACMCC), Bâtiment 701 CEA Orme des Merisiers, 91191 Gif-sur-Yvette Cedex, France
[4] Institut national de l'environnement industriel et des risques (INERIS), Parc Technol Alata, BP2, F-60550 Verneuil En Halatte, France

Correspondence to: L. Poulain (poulain@tropos.de)

**Abstract.** The Aerosol Chemical Speciation Monitor (ACSM) is nowadays widely used to identify and quantify the main component of fine particles in ambient air. As such, its deployment at observatory platforms is fully incorporated within the European Aerosol, Clouds and Trace Gases Research Infrastructure (ACTRIS). To ensure the consistency of the dataset, as well as instrumental performance and variability, regular intercomparisons are organized at the Aerosol Chemical Monitoring Calibration Center (ACMCC, part of the European Center for Aerosol Calibration, Paris, France). However, in-situ quality assurance remains a fundamental tracking point of the instrument's stability. Here, we present and discuss the main outputs of long-term quality assurance efforts achieved for ACSM measurements at the research station Melpitz (Germany) since 2012 onwards. In order to validate the ACSM measurements over the years and to characterize seasonal variations, nitrate, sulfate, ammonium, organic, and particle mass concentrations were systematically compared with collocated measurements including daily offline high-volume $PM_1$ and $PM_{2.5}$ filter samples. Mass closure analysis was made by comparing the total particle mass (PM) concentration obtained by adding the mass concentration of equivalent black carbon (eBC) from the Multi-Angle Absorption Photometer (MAAP) to the ACSM chemical composition, to that of $PM_1$ and $PM_{2.5}$ during filter weighting, as well as to the derived mass concentration of particle number size distribution measurements (PNSD). A combination of $PM_1$ and $PM_{2.5}$ filter samples helps identify the critical importance of the upper size cut-off of the ACSM during such exercises. The ACSM-MAAP-derived mass concentrations systematically deviate from the $PM_1$ samples when the mass concentration of the latter represents less than 60 % of $PM_{2.5}$, which is linked to the transmission efficiency of the aerodynamic lenses of the ACSM. The best correlations are obtained for sulfate (slope 0.96, $R^2 = 0.77$) and total PM (slope 1.02, $R^2 = 0.90$). Although, sulfate

does not exhibit a seasonal dependency, total PM mass concentration shows a small seasonal effect associated with an increase in non-water-soluble fractions. The nitrate suffers from a loss of ammonium nitrate during filter collection, and the contribution of organo-nitrate compounds to the ACSM nitrate signal make it difficult to directly compare the two methods. The contribution of m/z 44 ($f_{44}$) to the total organic mass concentration was used to convert the ACSM organic mass to OC by
using a similar approach as for the AMS. The resulting estimated $OC_{ACSM}$ was compared with the measured $OC_{PM1}$ (slope 0.74, $R^2$ = 0.77), indicating that the $f_{44}$ signal was relatively free of interferences during this period. The $PM_{2.5}$ filter samples use for the ACSM data quality might suffer from a systematic bias due to a size cutting effect as well as to the presence of chemical species that cannot be detected by the ACSM in coarse mode (e.g. sodium nitrate and sodium sulfate). This may lead to a systematic underestimation of the ACSM particle mass concentration and/or a positive artefact that artificially decreases
the discrepancies between the two methods. Consequently, ACSM data validation using $PM_{2.5}$ filters has to be interpreted with extreme care. The particle mass closure with the PNSD was satisfying (slope 0.77, $R^2$ = 0.90 over the entire period), with a slightly overestimation of the Mobility Particle Size Spectrometer (MPSS) derived mass concentration in winter. This seasonal variability was related to a change on the PNSD and a larger contribution of the super-µm particles in winter.

This long-term analysis between the ACSM and other collocated instruments confirms the robustness of the ACSM and its
suitability for long-term measurements. Particle mass closure with the PNSD is strongly recommended to ensure the stability of the ACSM. A near real-time mass closure procedure within the entire ACTRIS-ACSM network certainly represents an optimal way of both warranting the quality assurance of the ACSM measurements as well as identifying possible deviations in one of the two instruments.

**1. Introduction**

Aerosol particles strongly influence our environment, having especially an impact on the ecosystem and human health. In particular, fine particulate pollution directly affects mortality and morbidity (*e.g.* Gurjar et al., 2010;Ostro et al., 2007). Lelieveld et al. (2015) have estimated that air pollution, mostly < 2.5µm aerosol particles, may lead to 3.5 million premature deaths per year worldwide. Consequently, improving air quality represents a clear challenge, especially in urban areas.
Quantifying the impact of the regulations to the air quality and changes on aerosol chemical composition needs to perform continuous and long-term measurements of aerosol particle properties such as e.g. the particle number size distribution (PNSD) and the chemical composition. For this purpose, a European distributed facility of ground-based Aerosol Chemical Species Monitor (ACSM, Ng et al., 2011) is operated within ACTRIS (European Research Infrastructure for the observation of Aerosol, Clouds and Trace Gases, http://www.actris.eu). Complementary, the COST Action CA16109 Chemical On-Line
cOmpoSition and Source Apportionment of fine aerosol (COLOSSAL, https://www.costcolossal.eu) is gathering a wide community of European research groups (with even further international inputs, as well as participation of some regional air quality monitoring networks) interested in the fine aerosol fraction. One of the main objectives of these coordinated programs

is to investigate and understand the spatial variability of aerosol chemical composition on a continental scale, including temporal variability over days, seasons, and years. With such instrumental network, it is essential to keep a strong focus on the data quality as well as to assure that the results provided by each instrument are comparable to each other. Therefore, ACSM intercomparison workshops are regularly conducted within the framework of the European Center for Aerosol Calibration

(ECAC, www.actris-ecac.eu) at the Aerosol Chemical Monitor Calibration Center (ACMCC) in France. Data quality is ensured by determining instrumental variability between ACSMs (total mass 9 %, organic 19 %, nitrate 15 %, sulfate 28 %, ammonium 36 %, Crenn et al., 2015;Fröhlich et al., 2015a;Freney et al., 2019).

Although intercomparison exercises provide instrumental variability, a comparison between ACSM and collocated measurements remains a fundamental aspect of in-situ quality control. These intercomparisons are considered in a number of

publications (e.g. Fröhlich et al., 2015b;Petit et al., 2015;Parworth et al., 2015 ;Ovadnevaite et al., 2014 ;Ripoll et al., 2015;Minguillon et al., 2015;Poulain et al., 2011b;Poulain et al., 2011a;Huang et al., 2018;Takegawa et al., 2009;Wang et al., 2015 ;Crenn et al., 2015 ;Guo et al., 2015 ;Schlag et al., 2016 ;Sun et al., 2015). Usually, the comparisons between ACSM and collocated measurements were only performed for a few months up to one year. This might be perfectly adequate to ensure ACSM quality in that period. Only a few systematic comparisons with datasets longer than one year have been reported in the

literature (e.g. Fröhlich et al., 2015b;Petit et al., 2015;Parworth et al., 2015 ;Sun et al., 2015). Ovadnevaite et al. (2014) have written a rare published work that reports long-term AMS comparisons (3-years). Even though the authors successfully compared it with offline $PM_{2.5}$ filter samples, they did not discuss the variability of the correlation and potential source of uncertainties. However, it might not appear sufficiently long to properly evaluate the performance and stability of an instrument designed for long-term monitoring, *e.g.* covering periods of several years. Therefore, there is really a need for such year-long

investigations in order to evaluate the robustness of the instrument independently of calibrations and tuning as well as maintenance activities after technical failures (e.g. such as changing filament, pumps, etc.), seasonal variability, and properly define the limits of such exercises.

A key aspect of such a comparison is the individual upper size cut-off of each instrument. That of an ACSM (as well as the AMS since both are using the same aerodynamic lenses) is considered to be near-$PM_1$ (vacuum aerodynamic diameter),

regarding the approximate 30-40 % transmission efficiency of its aerodynamic lenses at 1 µm (Liu et al., 2007;Takegawa et al., 2009). Recently, a near-$PM_{2.5}$ aerodynamic lens has been developed (Xu et al., 2017). However, this new generation of instruments having a near-$PM_{2.5}$ cut-off are not within the focus of the present work. Overall, only a limited number of investigations referred to a direct comparison of the ACSM (as well as the AMS) with instruments that have a $PM_1$ cut-off. From those, multiple external references have been considered in order to compare individual species derived from offline

filter analysis (*e.g.* Ripoll et al., 2015;Minguillon et al., 2015;Poulain et al., 2011b;Poulain et al., 2011a;Huang et al., 2018), impactors (*e.g.* Takegawa et al., 2009;Wang et al., 2015), PILS (*e.g.* Crenn et al., 2015, Guo et al., 2015), and a MARGA (e.g. Schlag et al., 2016). Particle mass closure analysis has also been reported in the literature. It is achieved by adding equivalent Black Carbon mass concentrations (eBC) measured by an Absorption Photometer to the ACSM/AMS ones to obtain $PM_1$ mass concentrations and compare them with the ones derived from particle number size distributions (PNSD) measured by a MPSS

(Mobility Particle Size Spectrometer). One of the main difficulties of a comparison with the MPSS is volume to mass conversion, which requires the density of each detected species (*e.g.* Bougiatioti et al., 2016;Ortega et al., 2016;Ripoll et al., 2015). To avoid this, some studies have reported a direct comparison of mass concentration vs. volume concentration (*e.g.* Setyan et al., 2012;DeCarlo et al., 2008;Parworth et al., 2015;Huang et al., 2010). Although this second approach might represent an advantage in providing a direct estimation of the aerosol particle density, the absolute value of the resulting density might become difficult to interpret in some cases because of possible discrepancies between the two instruments types (e.g. Parworth et al., 2015). Although the MPSS is certainly the most popular instrument for particle mass closure analysis, the TEOM-FDMS can be used, since it provides the PM mass concentration directly (Petit et al., 2015;Guerrero et al., 2017).

The aim of the present work is to investigate the long-term stability and comparability between ACSM and collocated and well-established techniques over year-long measurements. Specific attention was put on the influence of the upper size cut-off diameter to better understand how it might affect the validation step and the robustness of the data. Finally, recommendations are provided for better on-site quality assurance and quality control of the ACSM results, which would be useful for either long-term monitoring or intensive campaigns.

## 2. Methodology

### 2.1 Research observatory Melpitz

The atmospheric aerosol measurements were performed at the TROPOS research station Melpitz (51.54 N, 12.93 E, 86 m a.s.l.), 50 km to the northeast of Leipzig, Germany. The station has been in operation since 1992 to examine the impact of atmospheric long-range transport on Central European background air quality (Spindler et al., 2012;Spindler et al., 2013). The site itself is situated on a meadow and is mainly surrounded by agricultural pastures and forests. The Melpitz observatory is part of EMEP (Co-operative Programme for monitoring and evaluation of the long-range transmissions of air pollutants in Europe, Level 3 station, Aas et al., 2012), ACTRIS, ACTRIS-2, GAW (Global Atmosphere Watch of the World Meteorological Organization), and GUAN (German Ultrafine Aerosol Network, Birmili et al., 2015;Birmili et al., 2009;Birmili et al., 2016).

All online instruments are set up in the same laboratory container and connected to the same air inlet. This inlet line consists of a $PM_{10}$ Anderson impactor located approximately 6 m above ground level and directly followed by an automatic aerosol diffusion dryer to actively keep the relative humidity on the sampling line below 40 % (Tuch et al., 2009). The aerosol flow is divided among a set of instruments by an isokinetic splitter (WMO/GAW, 2016) ensuring a representative sampling between the instruments. These instruments include a Multi-Angle Absorption Photometer (MAAP, model 5012, Thermo-Scientific, Petzold and Schönlinner, 2004) to measure the particle light absorption coefficients and the equivalent black carbon (eBC) mass concentration; a dual Mobility Particle Size Spectrometer (TROPOS-type T-MPSS; Birmili et al., 1999;Wiedensohler et al., 2012) to determine the PNSD from 3 to 800 nm (mobility diameter) alternating at ambient temperatures and behind a thermodenuder operating at 300 °C (Wehner et al., 2002); an Aerodynamic Particle Size Spectrometer (APSS; model TSI-

3321) to measure the PNSD from 0.8-10 μm (aerodynamic diameter), and a three wavelengths Integrating Nephelometer (model TSI-3563) for particle light scattering and backscattering coefficients.

For a basic overview of the physical and chemical aerosol characterization methods see e.g. Birmili et al. (2008);Spindler et al. (2012);Spindler et al. (2013);Poulain et al. (2014);Poulain et al. (2011b). Physical and optical aerosol instruments are frequently calibrated within the framework of the ECAC. The MPSS is calibrated at the WCCAP (World Calibration Center for Aerosol Physics), following the recommendations given in Wiedensohler et al. (2018). The PNSD uncertainty determined with the MPSS is approximately 10 %. The uncertainty of an APSS is between 10-30 %, depending on the size range (Pfeifer et al., 2016). The uncertainty of the MAAP is also within 10 % as determined by Müller et al. (2011).

## 2.2 ACSM

The ACSM (Ng et al., 2011) is connected to the same inlet of the previously described laboratory container. It is based on the same working principle as the widespread Aerodyne Aerosol Mass Spectrometer, AMS (Canagaratna et al., 2007;DeCarlo et al., 2006;Jayne et al., 2000). Compared to the AMS, the ACSM cannot provide size-resolved chemical information. It is equipped with a low-cost residual gas analyzer (RGA) type quadrupole (Pfeiffer Vacuum Prisma plus system) with a unit mass resolution instead of a time-of-flight mass spectrometer. The same aerodynamic lenses as in the AMS are also equipped in the ACSM, with a maximum transmission ranging from 75 to 650 nm, with ca. 30 to 40 % transmission efficiency at 1 μm (Liu et al., 2007). Consequently, the ACSM, like the AMS, provides the chemical composition of non-refractory near-$PM_1$ aerosol particles (organic, nitrate, sulfate, ammonium, and chloride) with a typical time-resolution of 30 min. The ACSM has been permanently operated at the Melpitz since June 2012. The present work will be, however, limited to the period from June 2012 to November 2017. The instrument was sent to the ACMCC (Aerosol Chemical Monitor Calibration Center) near Paris (France) twice to take part of the ECAC intercomparison workshops (Nov-Dec 2013, Crenn et al., 2015;Fröhlich et al., 2015a; and Mar-May 2016, Freney et al., 2019). Overall, the ACSM data capture 80 % of the time the instrument was deployed at Melpitz. Missing days correspond to either instrument failures or maintenance operations.

The ACSM was regularly calibrated according to the manufacturer's recommendations at that time with 350 nm monodispersed ammonium nitrate and ammonium sulfate particles selected by a DMA and using the jump scan approach. It is important to note that since recently, the recommended calibration method has changed to a full scan approach (Freney et al., 2019). The total particle number concentration was systematically set below 800 # cm$^{-3}$ to limit the artefact due to multiple charged particles. An overview of the ionization efficiency (IE) and relative ionization efficiency (RIE) for ammonium and sulfate can be found in Figure SI-1. On average, all performed calibration provides a mean IE value of 4.93 ($\pm$1.45) 10$^{-11}$ (mean $\pm$ std. dev.) and mean RIEs for ammonium and sulfate were 6.48$\pm$1.26 and 0.68$\pm$0.13, respectively. These values are very close to the ones used for the data evaluation as indicated in Figure SI-1. Overall, no clear trend for IE and RIE of sulfate can be observed over the period, while a small decrease in the RIE of ammonium can be reported. The lowest RIE of ammonium was

reported just after the replacement of the filament indicated a possible need for degassing and stabilization period. However, it is difficult to conclude if these tendencies could be associated with a possible aging effect of the instrument since it corresponds to a single instrument. Similar observations on various other individual ACSMs would be needed to allow for stating such a conclusion and a more systematic investigation of potential trends should then be performed with a large number of ACSM. The ACSM measurements and data analysis was made with the latest version of the Data Acquisition (DAQ) and Data Analysis (DAS) software's available at that time (Aerodyne, https://sites.google.com/site/ariacsm). The ACSM data was analyzed following the recommendation of manufacturer and applying a composition dependent collection efficiency (CDCE) correction based on the algorithms proposed by Middlebrook et al. (2012) to correct particle loss due to bouncing off the vaporizer before flash vaporization. It is important to note that the CDCE algorithm includes inorganic species only and did not consider a possible effect of the organics on the collection efficiency estimation.

## 2.3 Offline chemical characterization

Parallel to the ACSM, the high-volume samplers DIGITEL DHA-80 (Digitel Elektronic AG, Hegnau, Switzerland) collect particles with sizes selective $PM_{2.5}$ and $PM_{10}$ on preheated quartz fiber filters (105 °C) (Munktell, Type MK360, Sweden) for 24 hours from midnight to midnight. Samples were collected on a daily-based regime, whereas $PM_1$ was collected every 6 days. During some specific periods, related to different research projects that took place at the station, $PM_1$ sampling was also performed on a daily basis, as with $PM_{2.5}$ and $PM_{10}$.

After sampling, the filters were conditioned for 48 h at 20±2 °C and 50±5 % RH before being weighted by a microbalance Mettler-Toledo (AT 261). The filters were then extracted with ultrapure water (> 18 MΩcm) and analyzed through ion chromatography (ICS-3000, Dionex, USA) for water-soluble anions (column AS 18, eluent KOH) and cations (column CS 16, eluent methane sulfonic acid). For further descriptions of sampling and analyzing procedures, see Spindler et al. (2013).

For the chemical quantification of organic carbon (OC) and elemental carbon (EC), the sum of which is total Carbon (TC), a thermo-optical method was used. Rectangular punches (1.5 cm²) of every quartz filter were analyzed for OC and EC using the Lab OC-EC Aerosol Analyzer (Sunset Laboratory Inc. U.S.A.). The standard temperature protocol EUSAAR2 (Cavalli et al., 2010) was applied to distinguish OC and EC, and the transmittance mode was used for the charring correction. In European networks, like EMEP and ACTRIS, this thermos-optical method is the preferred technique for quartz fiber filters (final temperature 850 °C). Because filter samples were collected over 24 h, an artefact due to the evaporation of the most volatile compounds during warm periods, like ammonium nitrate or some organic, cannot be fully excluded (Schaap et al., 2004;Keck and Wittmaack, 2005).

**2.4 Air mass trajectory analysis**

A trajectory analysis was made based on 96 h backward trajectories for the altitude of 500 m above model ground with the NOAA Hybrid Single Particle Lagrangian Integrated Trajectory (HYSPLIT-4) model (Draxler and Hess (2004), http://www.ready.noaa.gov/ready/hysplit4.html). The trajectories were then analyzed using Zefir 3.7 (Petit et al., 2017) for the identification of potential aerosol sources using the Potential Source Contribution Function (PSCF). Because the filters were collected over 24 h, a total of 12 trajectories were considered for the analysis per day (i.e. every 2 hours), using the enlarge function of Zefir. Finally, the meteorological conditions as available from the HYSPLIT output for each trajectory calculation were also examined. Although backward trajectories were started at a height of 500 m, the Planetary Boundary Layer at the trajectory starting time could be at a lower altitude making the association between the ground-based measurements and the inflowing air mass difficult. Therefore, only air mass trajectories with a HYSPLIT-estimated PBL height above 500 m were further considered for analysis. Moreover, trajectories were cut off if they had a precipitation rate of over 1 mm h$^{-1}$ and an altitude of above 2000 m.

**3    Results**

To assure the data quality of the ACSM measurements, the results were systematically compared to *i*- daily offline filter samples (PM$_1$ and PM$_{2.5}$) of individual species (sulfate, nitrate, ammonium and organic) and *ii*- combined with eBC (MAAP) for mass closure analysis of both offline filter samplers and online MPSS. The accuracy of the comparison and the seasonal variabilities will be discussed in the following. All correlation fits were performed using least the orthogonal fitting approach without forcing it to zero.

**3.1 Comparison with offline chemical composition**

A comparison between total PM mass concentrations, sulfate, nitrate, and ammonium over the 5.5 years is plotted in Figure 1 for PM$_1$ and in Figure SI-2 for PM$_{2.5}$. The seasonal effect on the fitting's correlation to each species and PM cutting is presented in Figures 2 and SI-3 for PM$_1$ and PM$_{2.5}$, respectively. In the following, chloride will not be considered due to its very low concentrations and limited detection as described by Crenn et al. (2015). It is also important to note here that the comparison between ACSM and offline samplers generally consists of comparing dry aerosol online measurements to offline analyses of samples collected at ambient RH. A direct consequence is that the offline results might suffer from a cut-off shift due to aerosol hygroscopic growth when ambient RH is high (Chen et al., 2018). Based on this study, the cut-off shift due to aerosol hygroscopic growth should play a minor role at Melpitz, as this effect was estimated to influence the comparison by 2 % for marine air-mass and 1 % for continental air-mass. For European background stations, such a cut-off shift has been estimated to represent less than 10 % for PM$_1$ and 20 % for PM$_{2.5}$ particle mass loading, while it is stronger for marine or coastal stations

(up to 43 % for $PM_1$ and 62 % for $PM_{2.5}$). Therefore, such artefact has to be considered when comparison ACSM with offline measurements.

### 3.1.1 Sulfate

Over the entire period, the regression slope of the sulfate mass concentration comparison is suggesting overestimation of the ACSM compared to $PM_1$-filters (slope 1.45, $R^2 = 0.59$, Fig. 2 and Table SI-1). Better regression slopes were obtained in spring (slope = 0.98, $R^2 = 0.74$) and summer (slope = 0.87, $R^2 = 0.77$) than in fall (slope = 1.25, $R^2 = 0.58$) and winter (slope = 1.57, $R^2 = 0.61$). However, the overestimation observed throughout the entire period, seems to be strongly influenced by three periods taking place in January 2013, October 2015, and February 2017 (these periods are highlighted in Fig. 1). During these periods, the ACSM sulfate mass concentration strongly overestimates the $PM_1$ one. The correlations with the $PM_{2.5}$ sulfate mass concentration (Fig. SI-2 & SI-3) underline the systematic underestimation of the ACSM sulfate concentration throughout the entire period (slope 0.68, $R^2 = 0.85$), similar to the value reported by Petit et al. (2015) over 2 years of measurements in the region of Paris (France). This overestimation could be associated with the size-cutting difference between the two methods and the presence of not detected sulfate species on the coarse mode, such as sodium sulfate. The seasonal impact on the regression coefficients is less pronounced than in the comparison with $PM_1$, with regression slopes ranging from 0.64 ($R^2 = 0.85$) in spring to 0.94 ($R^2 = 0.85$) in summer. Contrary to the correlation with $PM_1$, no out-layers were identified here.

The following will focus on the ACSM sulfate's overestimation days. There are several reasons that might explain the sulfate overestimation by the ACSM. The first is a technical aspect, since the ACSM has a mass spectrometer with a unit mass resolution, it cannot distinguish between sulfate and organic fragments with the same m/z (for example, $C_6H_8^+$ and/or $C_5H_4O^+$ at m/z 80 for $SO_3^+$, or $C_6H_9^+$ and $C_5H_5O^+$ at m/z 81 for $HSO_3^+$), as already discussed in Budisulistiorini et al. (2014). Therefore, an increase of the organic signal at this m/z might lead to an overestimation of the ACSM sulfate mass concentration. Although our previous measurements using High-Resolution Time-of-Flight Aerosol Mass Spectrometer (HR-ToF-AMS) at Melpitz (Poulain et al., 2011b) support the presence of organic fragment on the UMR sulfate signal. The difference between the sulfate mass concentration based on UMR (as for ACSM) and the one obtained on the high-resolution (i.e. excluding the contribution of organic fragments on the sulfate signal) is below 10% indicating a minor impact of the organic to the sulfate signal. The second possible instrumental artefact is associated with the presence of a higher amount of organo-sulfate during these specific events. Indeed, organo-sulfate compounds lead to similar fragments as inorganic sulfate on AMS mass spectra (*e.g.* Farmer et al., 2010), which can contribute to the overestimation of the inorganic sulfate mass concentration. However, no particular change of $SO_3^+/SO^+$ and $HSO_3^+/SO^+$ ratios was observed when directly comparing their values before and after events, which can support neither the presence of organo-sulfate nor an increase of organic fragments at m/z 80 and 81. The second aspect is linked to sulfate size distribution. As can be seen in Figure SI-4, the $PM_1:PM_{2.5}$ ratio of the sulfate mass concentration has a pronounced season variability with a mean value of above 0.8 in spring and summer and of 0.6 in winter. The influence of

super-µm particles is also supported by the PNSD and PVSD as illustrated in Figure SI-5 for Feb. 2017, which coincides with the period with the highest discrepancy between the two methods (Fig. 2). In order to investigate a possible dependency on particle mass size distribution, a sensitivity test analysis was performed by investigating the changes of the fitting parameters parallel to the changes of the $PM_1$:$PM_{2.5}$ ratio on both sulfate and total PM mass concentrations (Fig. 3). In both cases, a clear change in regression slopes as well as intercept values could be observed whenever the $PM_1$:$PM_{2.5}$ ratio became smaller than 60 %. For days with a $PM_1$:$PM_{2.5} > 60$ %, the regression slope ranges from 0.82 and 0.97 with a small intercept value ranging from -0.06 to 0.015 µg m$^{-3}$. As soon as the $PM_1$ sulfate or the PM mass concentration represents less than 60 % of the $PM_{2.5}$, the ACSM overestimates the $PM_1$ sulfate. Therefore, the discrepancy between the ACSM and the $PM_1$ can be attributed to the individual upper size cutting of the two instruments, and it highlights the limits of such a comparison. As already mentioned, a minor effect of the RH to the cut-off shift of the offline samplers can be expected at Melpitz (Chen et al., 2018). Consequently, and for the following discussions on sulfate correlation, only the days with a $PM_1$:$PM_{2.5}$ ratio of above 60 % will be considered, which still covers more than 80 % of sampling days. The table SI-1 shows the fitting parameters obtained with and without considering the discussed size effect. The resulting correlation parameters show a regression slope of 0.96 (intercept = -0.06 and $R^2 = 0.77$, Fig. 2), which supports the results reported by Minguillon et al. (2015) (slope = 1.15) and Ripoll et al. (2015) (slope = 1.12). Seasons do not exercise a significant influence on the correlation between the two instruments, with regression slopes ranging from 0.85 in summer to 1.06 in fall, which supports the results reported by Budisulistiorini et al. (2014) and are better than the ACSM reproducibility uncertainties of 28 % reported by Crenn et al. (2015). The very low intercepts (-14 to 0 ng m$^{-3}$) might indicate a minor contribution of organo-sulfate on the ACSM sulfate (Fig. 2 and Table SI-1). As was already mentioned, the transmission efficiency of the aerodynamic lenses of the ACSM is decreasing from $\approx$ 600 nm ($d_{va}$) to 30-40 % at 1 µm. Consequently, the remaining transmission efficiency of the aerodynamic lenses above 1 µm influences the sulfate correlation with the $PM_1$ samples, leading to the reported overestimation of the ACSM sulfate mass concentration on days with a low $PM_1$:$PM_{2.5}$ ratio.

To investigate a possible origin of super-µm sulfate, trajectory analysis was performed for days that have a difference in sulfate mass concentrations in $PM_1$ and $PM_{2.5}$ that is larger than 1 µg m$^{-3}$ (i.e. Sulfate$_{PM2.5}$ − Sulfate$_{PM1}$ > 1 µg m$^{-3}$) (Fig.4). The trajectory density indicates that during these days, the air masses were dominated by two sectors (East and West), with the highest probability in a near Eastern area of Melpitz. This confirms the predominantly shallow Plenary Boundary Layer height (PBL) as calculated by HYSPLIT, which was below 500 m for approx. 90 % of the time (Fig. 4-c), rather indicating local/regional sources than resulting from long-range transport processes. For days that have a connection between calculated trajectories and measurements (e.g. PBL > 500 m), PSCF analysis identified super-µm sulfate located inside a narrow corridor starting from Melpitz and going East, then passing over the South of Poland (Fig. 4-b). Since this area is known to host several coal power plants, super-µm sulfate might be associated to coal emissions originating from this area.

### 3.1.2 Nitrate

The ACSM nitrate mass concentration tends to slightly overestimate the offline $PM_1$ nitrate throughout the entire period (slope = 1.16, $R^2$ = 0.80; Fig. 1 and 2). This overestimation is very small and within the error margin compared to massive overestimation in Ripoll et al. (2015) with a slope of 1.35 ($R^2$ = 0.77) and Minguillon et al. (2015) with a slope 2.8 ($R^2$ = 0.80).

A similar conclusion was also drawn by Schlag et al. (2016), during a comparison to MARGA $PM_1$ measurements. The overall results must be carefully interpreted since a strong seasonal effect has been observed (Fig. 2) with very poor correlation in summer (slope = 6.28, $R^2$ = 0.29) and a strong overestimation during the colder seasons (slope = 1.29, $R^2$ = 0.80). On the one hand, ambient temperature strongly influences the nitrate mass concentrations on filter samples. Ammonium nitrate is a semi-volatile compound that evaporates, leading to a loss of ammonium nitrate on the filter sample. In an intercomparison study of different sampling supports, Schaap et al. (2004) demonstrated that a quartz filter ($PM_{2.5}$ and $PM_{10}$) is a suitable material for sampling nitrate as long as temperature does not exceed 20 °C. The high-volume samplers are sitting outside and are not temperature controlled. Therefore, the inside temperature of the sampler is influenced by the outside temperature. This temperature artefact is clearly illustrated in Figure 5, when the variation of the ACSM:$PM_1$ nitrate ratio and the maximum temperature measured during the sampling day are compared. For ambient maximum temperatures above 10 °C, an increase of the ACSM:$PM_1$ ratio can be observed. Here it is imperative to note that the ambient maximum temperature did not reflect the temperature inside the sampler, solar radiation may also contribute to warm up the sampler. The highest discrepancy between the two methods corresponds to the warmest days, supporting the temperature artefact. Moreover, this also corresponds to the period with the lowest nitrate mass concentration measured by the ACSM (Fig. 5-b), which might also interfere with the absolute value of the ratio. On the other hand, the nitrate quantification by the ACSM is not free of artefacts. The ACSM's nitrate quantification is mainly based on the signals at m/z 30 ($NO^+$) and m/z 46 ($NO_2^+$), as well as on a minor contribution of $N^+$ and $HNO_3^+$ ions in a similar way as for the AMS (Allan et al., 2003). As with sulfate, interferences due to organic contributions at m/z 30 ($CH_2O^+$ and/or $C_2H_6^+$) and m/z 46 ($CH_2O_2^+$, $C_2H_6O^+$) also cannot be completely excluded. Because the ACSM is working at a unit mass resolution (UMR), it is not possible to distinguish nitrate from organic signals at these two m/z ratios. The direct consequence is a possible overestimation of the nitrate mass concentration in the UMR during high OA:$NO_3$ periods as shown by Fry et al. (2018). Another source of uncertainties concerning the ACSM nitrate mass concentration is the contribution of organo-nitrates to the nitrate signal, since the nitrate function of the organo-nitrate compounds fragments in a similar way to inorganic nitrate (Farmer et al., 2010). Therefore, the presence of organo-nitrate compounds artificially increases the ACSM-nitrate concentration. Kiendler-Scharr et al. (2016) have already shown that organo-nitrate compounds contribute to a significant fraction of the default AMS-$NO_3$ signal, especially in summer. It represents 57 % and 29 % of the default nitrate measured by an AMS at Melpitz in summer and winter, respectively (Kiendler-Scharr et al., 2016). Since the ACSM and the AMS are based on a similar principle, a simple assumption was made to tentatively correct the ACSM nitrate assuming the following: Firstly, the winter nitrate filter-$PM_1$ mass concentration is free of temperature artefacts, and secondly, the contribution of the organo-nitrate to the ACSM nitrate signal is being constant

(29 %) over winter and years as previously reported for winter AMS measurements at the site. The resulting winter nitrate mass concentration has a better correlation to the filter-$PM_1$ (slope 0.88, $R^2 = 0.77$, Fig. SI-6). This indirectly confirming the importance of organo-nitrate contributions to the default ACSM nitrate mass concentration during wintertime. Therefore, one should be careful when comparing the ACSM nitrate with an offline system because of both temperature and organo-nitrate

artefacts. Comparing the ACSM with a $PM_1$ MARGA for a year, Schlag et al. (2016) have obtained a $R^2$ of 0.96 throughout the year, without discussing seasonal variability. Consequently, all these results tend to indicate that the ACSM inorganic nitrate should properly correlate with the temperature artefact-free $PM_1$ nitrate measurements, as can be achieved by a PILS or a MARGA for example. Moreover, calculating the difference of nitrate mass concentrations between the ACSM and an online $PM_1$ system (e.g. PILS or MARGA) might represent a possible way to estimate the organo-nitrate concentration as

reported by Xu et al. (2015) using HR-ToF-AMS vs. PILS or by Schlag et al. (2016) using ACSM and MARGA. Due to the unit mass resolution of the ACSM, direct quantification of particulate organo-nitrate remains a challenging task and more investigations are needed to better understand how organo-nitrate can be detected by the ACSM.

In a first approach, comparisons with the $PM_{2.5}$ nitrate mass concentration provided better correlation coefficients over the

entire period (slope = 0.76, $R^2 = 0.77$), as well as in winter (slope = 0.74, $R^2 = 0.69$), spring (slope = 0.77, $R^2 = 0.83$), and fall (slope = 0.96, $R^2 = 0.74$), compared to $PM_1$ (Fig. SI-2 and SI-3). Similar to $PM_1$, no correlation was found in summer. Here, the temperature effect on the filters as well as on organo-nitrate artefacts seems to have a less pronounced influence. Consequently, the presence of non-volatile nitrate compounds such as sodium nitrate ($NaNO_3$), resulting from the reaction of marine sodium chloride with $HNO_3$ when marine air masses cross polluted areas (Finlayson-Pitts and Pitts, 1986;Pio and

Lopes, 1998), might explain the difference of the correlations between $PM_1$ and $PM_{2.5}$. This is supported by the absence of significant effects of the $PM_1:PM_{2.5}$ nitrate ratio to the fitting parameters when comparing the ACSM nitrate with the $PM_1$ (Fig. 3). The influence of sodium nitrate at Melpitz has already been discussed in Stieger et al. (2017), comparing $PM_{10}$ MARGA results with ACSM ones throughout the same period. Consequently, comparisons between the ACSM and $PM_{2.5}$ nitrate measurements could be strongly biased by coarse mode sodium nitrate that cannot be detected by the ACSM. This

might be an important source of artefact, especially for sites under the influence of processed marine air masses, and might lead to a wrong validation of the ACSM nitrate measurements.

### 3.1.3 Ammonium

The ammonium mass concentration measured by the ACSM mostly corresponds to ammonium nitrate and ammonium sulfate

salts. Before comparing ACSM and offline $PM_1$ ammonium mass concentration, the neutralization state of the particles was estimated for both datasets assuming a full neutralization by nitrate, sulfate, and chloride as described in e.g. Sun et al. (2010). In both approaches, particles can be considered as fully neutralized during the entire period with no seasonality (Fig. SI-7) in agreement with previous AMS measurements made at the same place (Poulain et al., 2011b). Correlations with offline systems

fall somewhere between the two previously discussed ions. During the cold season, the ACSM ammonium mass concentration matches the $PM_1$ (slope 1.02, $R^2 = 0.83$), which supports the larger fraction of ammonium nitrate in the total PM as well as the size effect of sulfate during wintertime (Fig. 1 and Fig. 2). During the warm season, the evaporation of ammonium nitrate as discussed before will also induce a loss of ammonium on the filter samples compared to the online measurements leading to an under-estimation of the ammonium concentration on the offline sampler as well as a poor correlation ($R^2 = 0.49$). Similar conclusions can also be drawn when comparing it to the $PM_{2.5}$ ammonium mass concentration (Fig. SI-2 & SI-3).

### 3.1.4 OM and OC

The ACSM provides organic aerosol (OA) mass concentrations but contrary to the inorganic species no direct comparison with collocated organic mass measurements provided organic mass concentration is possible. Actually, only ACSM or AMS systems are nowadays able to provide such measurements and other methods - primarily based on the thermal and/or optical properties of carbonaceous aerosols - are estimating organic carbon (OC) mass concentration instead of OA. Here, offline OC measurements are available from the thermal-optical analyses of filter punches, allowing for comparing both parameters over the entire period of the study. In the following, the limitations of both methods are discussed. First of all, the organic aerosol mass concentration is defined as the sum of the non-attributed inorganic species fragments from the aerosol mass spectra as defined by Allan et al. (2004). A wrong assignment or correction of the fragmentation table during the data analysis process could be a source of mis-quantification of the organic mass concentration. For example, the fragment $CO_2^+$ (m/z 44) is the major signal on the organic mass spectra. It can suffer from substantial measurement biases, i.e., the so-called Pieber effect (Pieber et al., 2016;Freney et al., 2019) associated with interference due to nitrate signal. This artefact can lead to an overestimation of the m/z 44 and consequently directly affects the total organic mass concentration. Unfortunately, a thorough quantification of this effect on the present dataset is not possible, as the relevant method to do so includes regular full scan calibrations which has been proposed only recently, and further works are still needed to define associated correction procedures (Freney et al., 2019). Another main source of uncertainty for OA concentration estimates is linked to the assumption of a constant RIE. Here, it has been set at its 1.4 default value during the whole period of the study while it is known that RIE-organic and/or its CE can be influenced by the chemical composition of the organic (Xu et al., 2018). As already mentioned, organic is not included in the CDCE estimation method from Middlebrook et al. (2012), which might also have a potential impact on the resulting mass concentration. Overall, an uncertainty of 19 % in the ACSM organic mass concentration can be considered based on the ACSM reproducibility analysis made by Crenn et al. (2015).

OC mass concentrations derived from the offline analyses of filter samples are also subject to measurement uncertainties They are obtained according to a specific method (here the EUSAAR2 thermal-optical protocol). Applying another method will directly influence the OC concentration (Cavalli et al., 2010;Zanatta et al., 2016;Chiappini et al., 2014). Moreover, the samplers used for this study were sitting outside and were not temperature controlled. A direct consequence is that the evaporation of

the more semi-volatile organic during warm days must be expected, which similarly impacted the measured OC concentration than for ammonium nitrate discussed above.

Keeping in mind all the mentioned uncertainties on each method, the OA mass concentration was compared to the offline OC mass concentration, which can therefore be considered as a fair estimation of the OM:OC ratio (Fig. 6-a). Correlation between OA and OC is not significantly impacted by the $PM_1$:$PM_{2.5}$ threshold ratio of 0.6 as for inorganics (Table SI-1). This supports the fact that organic is mainly distributed on the sub-μm size range throughout the year (Fig. SI-4). As expected, a lower OM:OC ratio was obtained in winter (slope = 1.29, $R^2$ = 0.78), which corresponds with the period with the largest anthropogenic influence. The highest OM:OC ratio was obtained in summer (slope = 2.74, $R^2$ = 0.68), corresponding with the SOA formation maximum. Although such a seasonal variation is coherent with a priori expectations (notably considering higher SOA contribution at summertime), biases related to instrumental uncertainties should still be considered. In a similar way than for nitrate, ambient temperature affects the OC leading to a systematic extreme OM:OC ratio during summer (Fig. 6-c). Consequently, the summer's slope of 2.74 is certainly overestimated. However, some extreme values are found also for some winter days, which can therefore not be associated with a temperature artefact on the offline samplers. Such wintertime discrepancies might rather be attributed to the above-mentioned ACSM uncertainties related to RIE for organics, CE estimation and/or substantial influence of the so-called Pieber effect.

Another way to compare ACSM measurements to OC concentrations could be envisaged based on some previous works using AMS systems. Indeed, the estimation of the OM:OC ratio from AMS measurements is normally not done on a direct comparison of organic particle mass concentrations with collocated OC measurements but rather estimated based on the elemental analysis of the high-resolution organic mass spectra Aiken et al. (2007) and Aiken et al. (2008) or the variability of the $f_{44}$, the contribution of mass m/z 44 (mostly $CO_2^+$) to the total organic signal when only unit mass resolution mass spectra are available (Aitken et al., 2008, Ng et al., 2010). Both methods were reinvestigated and improved by Canagaratna et al. (2015) providing the following equations to convert the $f_{44}$ signal of an AMS into O:C and OM:OC ratios:

$$O:C = 0.079 + 4.31 \times f_{44} \tag{1}$$

$$OM:OC = 1.29 \times O:C + 1.17 \tag{2}$$

By a systematic comparison of the two approaches, the elemental analysis, and the $f_{44}$, Canagaratna et al. (2015) concluded to an accuracy of 13 % of the $f_{44}$ proxy for SOA traces decreasing for primary OA standards having an $f_{44}$ < 4 % on average. Considering that m/z 44 is systematically the dominate fragment of the organic mass spectra for ambient measurements and that the ACSM is based on similar principle than the AMS, it is relevant to apply the $f_{44}$ approach on the ACSM organic results as a proxy for ambient OC, and compare the results with the well-established offline OC method. Therefore, equations 1 and 2 were applied in the present dataset to estimate OC mass concentrations from the measured ACSM organic mass concentration

($OC_{ACSM}$) and to compare them to the OC-PM$_1$ (for the entire dataset: slope = 0.65, R² = 0.73, Fig. 7-a). As previously shown, a seasonal trend can also be observed here, with a unity regression slope obtained during summer periods (slope = 0.99, R² = 0.64), whereas a lower slope (0.56, R² = 0.82) was obtained in winter (Fig. 7-a). Here, the different instrumental and technical uncertainties have to be considered. Contrary to nitrate, temperature seems to have a less significant impact on the ratio between

the $OC_{ACSM}$ and the $OC_{PM1}$, as can be seen in Figure SI-8. However, the extreme $OC_{ACSM}$:$OC_{PM1}$ ratio values mostly happened during warm days supporting our previous conclusion on the temperature artifact on the $OM_{ACSM}$:$OC_{PM1}$ ratio. As was mentioned above, the extreme ratio values during winter might result from a possible variability of the organic RIE as well as a possible co-call Pieber effect on the m/z 44 that directly affects the estimation of the $OC_{ACSM}$. Despite this agreement between ACSMs, Crenn et al. (2015) showed a large variability concerning the $f_{44}$ signal itself during the ACSM intercomparison

exercise. This variability was attributed to an instrument-dependent difference in the vaporization conditions. For this reason, the authors did not recommend to systematically use the $f_{44}$ approach to estimate the O:C ratio, as it can be achieved with the AMS and done here, or to interpret the resulting O:C ratios with caution. Since the $OC_{ACMS}$ results are well supported by the offline analysis, we can conclude that our ACSM provides a relatively realistic value of the $f_{44}$ over the considered timeframe and consequently, a reasonable proxy for the OM:OC ratio. However, we cannot rule out that a similar approach would provide

the same results when using another ACSM at Melpitz and/or when applying the present method in another location. Further systematic comparisons between the ACSM and collocated OC-PM$_1$ measurements should be performed in order to better investigate and characterize the suspected instrument vaporization dependency and/or a possible matrix effect depending on the dominant type of aerosol chemical composition at the considered sampling site, which might influence both the $CO_2^+$ signal and the organic RIE.

Comparison of the $OC_{ACSM}$ with the OC PM$_{2.5}$ (Fig. 7-b) presents a systematic underprediction of the ACSM organic, which can be directly related to the size distribution of organic carbon between PM$_1$ and PM$_{2.5}$ (Fig. SI-4). Similar seasonality effects can be observed, which matches the quite constant distribution of the OC between PM$_1$ and PM$_{2.5}$ over the course of a year.

**3.2 Mass closure analysis**

Before performing a mass closure analysis, the total ACSM particle mass concentration (*i.e.* the sum of organic, nitrate, sulfate, ammonium, and chloride mass concentrations) was completed by adding the eBC PM$_1$ mass concentration. The eBC (PM$_{10}$) measured by the MAAP was converted to PM$_1$ by using a factor 0.9, which was obtained by running two MAAPs at Melpitz side by side with different inlets, see Poulain et al. (2011b). The resulting total PM$_1$ mass, later referred to as the ACSM-

MAAP-derived mass concentration, was then compared to the particle mass concentration obtained by weighting filters (PM$_1$ and PM$_{2.5}$) as well as to the calculated particle volume and mass concentration from the PNSD of the MPSS.

### 3.2.1 Mass closure with offline filters

In a similar way to sulfate, the mass closure between the online ACSM-MAAP-derived mass concentration and the offline $PM_1$ mass concentrations strongly depends on the $PM_1:PM_{2.5}$ ratio (Fig. 3). Because nitrate and organic did not present such a dependency to the $PM_1:PM_{2.5}$ ratio (Fig. 3), the sulfate size distribution should be the main driver of the total mass correlation.

Consequently, the same $PM_1:PM_{2.5}$ threshold of 0.6 is applied in the following for the mass closure analysis and its discussion. It leads to a strong consistency between the online and the offline methods (slope = 1.02, $R^2$ = 0.90, Fig. 2 and table SI-1). Our results support those of Petit et al. (2015), who use a $PM_1$ TEOM-FDMS for mass closure over a 2 years' timeframe (slope 1.06). However, Guerrero et al. (2017) stated that a regression slope of 0.81 could also be found in the $PM_1$ TEOM-FDMS. Therefore, it is not possible to conclude whether this difference in correlation results between the two studies depends on the

location or the presence of more coarse mode. Moreover, a possible loss of the more volatile compounds during the heated transmission line of the TEOM-FDMS could also occur.

Looking at the different seasons, the regression slopes were always around unit except in fall (slope = 1.31), the overestimation of which will be discussed in the following. Despite a near unity regression slope of 0.96 in summer, the low $R^2$ and the high intercept value (-3.59 µg m$^{-3}$) both suggest a possible bias between the two methods. Chemical analysis performed on the filter

samples cannot typically account for the entire mass, leading to the so-call residual mass fraction. This residual mass fraction is made out of all the non-water-soluble compounds such as mineral dust, carbonated or metal ones that are not detected. Here, the residual mass fraction was calculated as the difference between the weighted filter mass and the sum of the detected compounds (Fig. SI-9). It is important to note here, that to properly convert the OC into OM and to consider all the different limitations inherent to both online and offline approaches, the seasonal means OM:OC ratio values (Fig. 6-b) were applied.

Figure 8 illustrates how this residual mass fraction interferes with the comparison of the ACSM-MAAP-derived mass concentrations. In summer, the residual mass fraction represents a significant part of the $PM_1$ mass concentration (above 60 %), explaining the low correlation coefficient and the large intercept value in this season. Similar conclusions can be drawn for fall. The increase of residual mass fraction in summer and fall could be associated with a larger resuspension of crustal material on dry and warm days and/or with agricultural activities (e.g. plowing) at these times of the year. Since mineral dust is not

detectable by the ACSM, the presence of such compounds in the $PM_1$ could significantly influence mass closure results and must, therefore, be considered in such an approach.

A comparison with the $PM_{2.5}$ mass concentration provides a regression slope of 0.69 ($R^2$ = 0.77, Fig. SI-3), which matches the comparisons from the literature using $PM_{2.5}$ TEOM-FDMS mass concentration (*e.g.* Sun et al., 2015;Sun et al., 2012). A

seasonal effect on the correlation can be observed (Fig. SI-3). In winter, the discrepancy between online and offline techniques becomes more pronounced (slope = 0.65, $R^2$ = 0.88). This supports the seasonal variation of the $PM_1:PM_{2.5}$ ratio (Fig. SI-5) as well as the impact of coarse mode sulfate that was previously mentioned. Similar results were also shown by Sun et al. (2015) when performing mass closure with a $PM_{2.5}$ TEOM.

### 3.2.2 Mass closure with PNSD

The PNSD has been continuously measured in parallel to the aerosol mass spectrometer and can, therefore, be used to perform mass closure analysis between ACSM-MAAP and PNSD (ranging from 10 to 800 nm, mobility diameter). To ensure a robust comparison between the two systems, two approaches are reported in the literature: the first one consists of converting the ACSM-MAAP mass concentration into volume and the PNSD in volume concentration. The second one consists of converting the PNSD into mass concentration. Both approaches are based on the same assumptions of (i) spherical, (ii) fully internally mixed particles, and (iii) an identical chemical composition over the entire size distribution to estimate a chemical time-dependent gravimetric particle density based on the following equation from Salcedo et al. (2006):

$$density = \frac{[Total_{AMS} + eBC]}{\frac{[NO_3^-] + [SO_4^{2-}] + [NH_4^+]}{1.75} + \frac{[Cl^-]}{1.52} + \frac{[Org]}{1.2} + \frac{[eBC]}{1.77}} \tag{1}$$

Here, the density was assumed to be 1.75 g cm$^{-3}$ for ammonium nitrate and ammonium sulfate (Lide, 1991), 1.52 g cm$^{-3}$ for ammonium chloride (Lide, 1991), and 1.2 g cm$^{-3}$ for organic matter (Turpin and Lim, 2001). Finally, a density of 1.77 g cm$^{-3}$ (Park et al., 2004) was applied for eBC. A discussion of eBC density can be found in Poulain et al. (2014).

It is important to note that for the volume concentration approach, both measurements (ACSM-MAAP and MPSS) remain independent between each other, which is not the case when using mass concentration. However, only a few numbers of papers reported a comparison between AMS or ACSM and MPSS in volume concentration (e.g. DeCarlo et al., 2008;Elsasser et al., 2012). Even though the two variables are non-independent on the mass concentration approach, it remains the most commonly used. A possible reason is that the mass concentration unit remains easier to use and interpreted as the volume concentration since atmospheric measurements are usually made in mass concentration.

Here, we investigated comparison results obtained using each of these approaches. Results are summarized in Figure 9-a for the volume concentration approach and Figure 9-b for the mass concentration one. Over 5.5 years of measurements, the ACSM-MAAP-derived volume or mass concentration correlates well with the estimated volume or mass concentration of the MPSS with similar slopes of 0.79 and 0.77, respectively ($R^2 = 0.90$, Fig. 9-a and 9-b). This matches similar previous comparisons at the same place with an AMS (Poulain et al., 2014). Therefore, the selected method (volume or mass) did not substantially influence the comparison results. In the conditions of the present study both approaches could be applied for the station of Melpitz. Since comparison in mass concentration is the more commonly used, we will focus on it in the following discussions."
 However, our results also highlight a non-negligible seasonality effect on mass closure, with a better slope in warmer seasons (summer, slope 0.92, $R^2 = 0.85$) than in cold ones (winter, slope 0.75, $R^2 = 0.91$). A similar seasonality was already reported by Fröhlich et al. (2015b) using a ToF-ACSM at the Jungfraujoch (Switzerland) during 14-month measurements. The median particle number (Fig. 9-b) and volume (Fig. 9-c) size distributions throughout the winter and summer months emphasize two

different behaviors. In winter, the fine mode volume distribution peak occurs around 340 nm, while in summer it's around 250 nm. Moreover, the particle volume size distribution in winter also shows a higher concentration of the largest size bins. This difference corresponds to the higher concentration of super-µm particles in winter as confirmed by the seasonality of the $PM_1:PM_{2.5}$ mass ratio (winter 0.73, summer 0.84, Fig. SI-4). The PNSD provided by the MPSS is corrected from multiple-charged particles artefact in the sub-µm size range, and in case of low contributions of super-µm particles, the multiple-charged particles coming from super-µm particles on the PNSD are negligible. However, in case of a large coarse mode concentration, multiple-charged particles from the super-µm size range might also affect sub-µm size distribution, leading to an overestimation of the PNSD. This interference represents a possible source of artefact for the MPSS in such a case (Birmili et al., 2008). This artefact may plausibly explain the seasonality of the mass closure. An extended particle number size distribution by merging the MPSS and the APSS is presented in Figure SI-5 for February 2017 to illustrate the impact of super-µm particle on size distribution. This period was strongly influenced by coarse mode particles that interfered with the comparison between the ACSM and offline sulfate and PM as discussed earlier. On the other hand, and as previously mentioned, the transmission efficiency of the aerodynamic lenses of the ACSM decreases to about 30-40 % from $\approx$ 650 nm ($d_{va}$) to 1 µm. Consequently, the ACSM certainly underestimates the particle mass concentration for the larger size bins compared to the MPSS, which might also have a significant effect on particle mass closure with the MPSS in wintertime.

Assuming spherical particles and a constant density of 1.6, the size cutting of the Digitel $PM_1$ corresponds to a volume equivalent diameter of approx. 790 nm, which is quite similar to the MPSS (800 nm). The comparison between the MPSS-derived mass concentration and the $PM_1$ filter one also supports our conclusions (Fig. 10). Whereas the correlation slope of 1.79 ($R^2$ = 0.75) over the entire dataset seems to indicate an overestimation of the offline $PM_1$ compared to the MPSS, the correlation slope is strongly influenced by some winter days. Here, again, the discrepancy between the two methods can be linked directly to the upper size cut of each system. As shown in Figure 10, the discrepancy between the $PM_1$ and the MPSS-derived mass concentration is always associated with days with a low $PM_1:PM_{2.5}$ ratio corresponding to a larger contribution of the coarse mode particle compared to the other days. This result confirms the individual size cutting effect as well as supports our conclusions on a non-negligible artefact of super-µm multiple charge particles on the estimated MPSS mass concentration on specific winter days. During summertime, the $PM_1$ filter mass concentrations underestimate those derived from the MPSS, which have to be associated with the already discussed, loss of semi-volatile compounds on the filters.

## 4    Summary and conclusion

A systematic comparison between the ACSM and collocated measurements (including daily $PM_1$, $PM_{2.5}$, and MPSS) over a period of more than 5 years was performed to investigate the robustness of the ACSM as well as to identify the limits of such an exercise and the possible sources of uncertainties and artefacts. For such an exercise, it is fundamental to ensure isokinetic

flow splitting between the different instruments connected to the main sampling line to ensure a homogeneous distribution of the air sample.

The comparison with the offline daily $PM_1$ samples over the entire period highlights a strong artefact due to the presence of super-µm sulfate. This artefact becomes non-negligible as soon as the $PM_1$:$PM_{2.5}$ ratio of the sulfate (and subsequently the

total PM mass concentration) is below 60 %. The differences were directly associated with the specific size cutting of each instrument and the effect of the remaining transmission efficiency of the aerodynamic lenses of the ACSM above 1 µm. Moreover, similar conclusions were also drawn for the mass closure between the MPSS and $PM_1$ mass concentrations, confirming individual instrumental upper size cut-off effect. Because this artefact strongly depends on the size distribution of sulfate salts, it certainly depends on the sampling location and the origin of the different aerosol sources. Moreover, this effect

should also depend on the aerodynamic lenses itself, which should not all have exactly the same transmission efficiency about 1 µm, leading to a certain instrument dependency. Considering these instrumental limits, the ACSM sulfate mass concentration strongly correlates with the one measured on the filters without any pronounced seasonal effect (slope: 0.96, $R^2 = 0.77$). This also indicates a minor contribution of organo-sulfates to the ACMS sulfate mass concentration at the measurement's site. Consequently, the $SO_4$-$PM_1$ appears to be a crucial parameter to ensure the $SO_4$-ACSM validation as well as to support the

ACSM's sulfate calibration. In contrast, nitrate mass closure suffers from strong sampling artefacts for both instruments. On the one hand, offline measurements are strongly affected by a temperature effect, leading to the evaporation and loss of ammonium nitrate as was observed for maximum day temperatures of above 10 °C. On the other hand, organo-nitrate compounds lead to a systematic over-prediction of nitrate by the ACSM, which was clearly demonstrated in winter. Therefore, more investigations on the quantification of organo-nitrate by the ACSM are required in order to reduce this uncertainty.

The ACSM organic mass concentration correlates with the OC-$PM_1$ ($R^2 = 0.68$ to 0.81), supporting the ACSM organic measurements. The regression slopes have a clear seasonal variability that matches the expected change of the oxidation state of organic throughout the year. Despite the large inter-instrumental variability of the $f_{44}$ reported by Crenn et al. (2015), the $f_{44}$ was used to convert the organic mass measured by the ACSM into OC by applying the method proposed by Canagaratna et al. (2015), which was developed for the AMS. The good match between the OC-ACSM and OC-$PM_1$ (slope ranging from 0.99

in summer to 0.56 in winter with an overall value of 0.65) confirmed that the approach for this instrument and at this sampling place, is also suitable for the ACSM. Nevertheless, the method might be difficult to apply for short time measurements (e.g. a few weeks only), where low/high extreme ratios may be misinterpreted, and results interpreted with cautions, such OA-OC comparison and $OC_{ACSM}$:$OC_{offline}$ methods shall then preferably be used on long term continuous measurements. Finally, it should also be noted that OC is the only regulated organic aerosol-related variable commonly monitored within current air

quality networks (Directive 2008/50/CE, 2008;WMO/GAW, 2016) whereas equivalent methods for a better OA quantification at high-time resolution are still to be standardized, reinforcing the need for much more systematic comparison exercises at various locations.

Not surprisingly, the comparison to the offline $PM_{2.5}$ first highlights the importance of the size cut-off of the filter samples. This is true for all considered species (PM, nitrate, sulfate, ammonium, and organic). Although such conclusions might appear

quite trivial, the ACSM as well as the AMS are often compared to $PM_{2.5}$ filters. This is certainly the case, because $PM_{2.5}$ is the monitoring standard of air quality in several countries like the USA, Canada, and China, contrary to $PM_1$. Therefore, for such a comparison, the limitations due to the different size cuttings must be considered. Moreover, possible cut-off shift due to ambient relative humidity effect on the offline measurements could represent a non-negligible parameter and has to be considered during such an exercise, especially for marine stations.

The total $PM_1$ mass balance between online (ACSM and MAAP) and offline $PM_1$ matches throughout the entire time period (slope: 1.02, $R^2$ 0.90) as well as the different seasons when considering the size effect mentioned before. However, non-water-soluble species like dust, metals and carbonate that were not analyzed in the filter samples in this study, and which are also not detected by the ACSM, influence the correlation especially in summer, leading to a lower correlation coefficient during this period ($R^2 = 0.40$). Mass closure with the PNSD certainly represents the best way for in-situ quality control as well as for tracking a possible drift on the ACSM performance. It can be performed by converting the ACSM-MAAP mass concentration into volume concentration or by converted the MPSS volume concentration into mass concentration both using time-dependent density and assuming spherical and fully internally mixed particles. The volume approach is the most robust since it enables a strictly independent method. Being more interpretable, the mass approach may be used instead of for error quantification as long as it agrees with the volume approach. For the present dataset, the selected method did not substantially influence the comparison results. Compared to offline samples, comparisons with the MPSS do not only have a quite stable correlation over the years and the seasons, but the mass closure between the ACSM and MPSS also presents the main advantage to be done at a near real-time approach, since no further laboratory analyses are needed. Consequently, near real-time mass closure between the ACSM and MPSS should be considered in the near future as a standard way for in-situ quality control of measurements. Moreover, this approach does not remain free of artefacts related to the instrumental upper size cut-off diameter. This should be considered for sampling places with an important coarse mode fraction, in order to considered artefacts induced by both the remaining aerodynamic lens transmission efficiency of the ACSM and the contribution of multiple-charged particles from coarse mode on the PNSD spectra.

Finally, our results clearly emphasize the different limits of a comparison to collocated instruments and the effects of each individual instrumental upper size cut-off diameter. Consequently, there is a need for a better and systematic characterization of the transmission efficiency of the aerodynamic lenses of the ACSM on the upper size range. This knowledge will also certainly be useful to better understand the instrumental variability. Nevertheless, such near real-time comparisons certainly represent the best way to ensure long-term quality assurances of the ACSM measurements, especially at a station where the ACSM is used for long-term monitoring of particle chemical composition. More systematic comparisons performed in a similar way as in the present work over a long time-period in different environments as well as using different reference methods (e.g. TEOM-FDMS, beta-gauge or a PILS with $PM_1$ inlet for example) are still needed to better characterize the robustness of the ACSM over a long sampling time.

*Data availability*: all data is available upon request to the corresponding author and are deposited on the EBAS-NILU database (http://ebas.nilu.no/Default.aspx).

*Authors contributions*: LP, TT, AG, BS collected the data, LP performed data analysis on the ACSM, GS contributed to the evaluation of the offline dataset and TT to the MPSS. All co-authors participated to the interpretation of the results. LP lead the writing of the manuscript to which authors contributed.

*Competing of interest*: the authors declare that they have no conflict of interest.

**Acknowledgments**

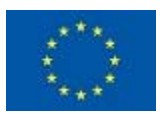 This project has received funding from the European Union's Horizon 2020 research and innovation programme within the infrastructure projects ACTRIS (Aerosols, Clouds, and Trace gases Research InfraStructure Network; EU FP7 grant agreement N°262254) and ACTRIS-2 (Aerosols, Clouds, and Trace

gases Research InfraStructure, grant agreement N°654109). This work was also supported by the COST Action COLOSSAL CA16109. The physical measurements were also funded by the German Ultrafine Aerosol Network GUAN, which was jointly established with help of the German Federal Environment Ministry (BMU) grants F&E 370343200 (German title: "Erfassung der Zahl feiner und ultrafeiner Partikel in der Außenluft"), 2008–2010, and F&E 371143232 (German title: "Trendanalysen gesundheitsgefährdender Fein- und Ultrafeinstaubfraktionen unter Nutzung der im German Ultrafine Aerosol Network

(GUAN) ermittelten Immissionsdaten durch Fortführung und Interpretation der Messreihen") 2012–2014. We also acknowledge the WCCAP (World Calibration Center for Aerosol Physics) as part of the WMO-GAW program. The WCCAP is base-funded by the German Federal Environmental Agency (Umweltbundesamt), Germany. Support by the European Regional Development Funds (EFRE – Europe funds Saxony) is gratefully acknowledged. The authors also thanks R. Rabe especially for technical support in the field and A. Rödger, A. Dietze, and S.Fuchs for numerous laboratory analyses.

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

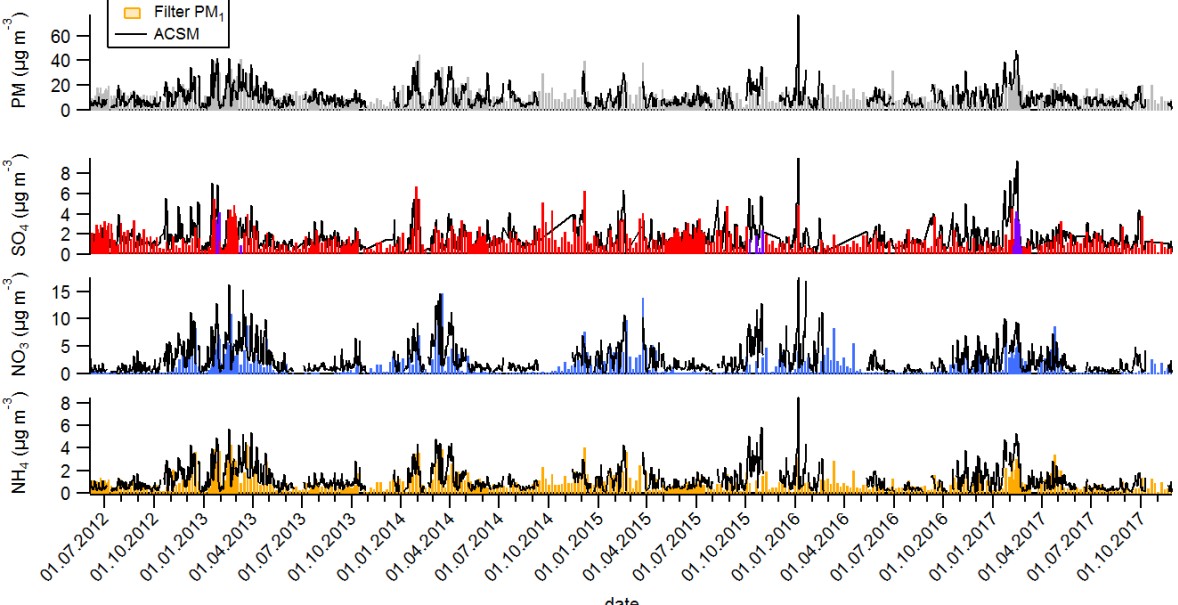

**Figure 1: Time series ACSM (daily averaged, black line) and 24 h PM$_1$ filter samples (colored bars) for the total particle mass concentration, the mass concentration of sulfate, nitrate, and ammonium. The particulate matter (PM) corresponds with the sum of ACSM species and eBC$_{PM1}$ for the online instrument and the PM$_1$ filter mass for the offline samples.**

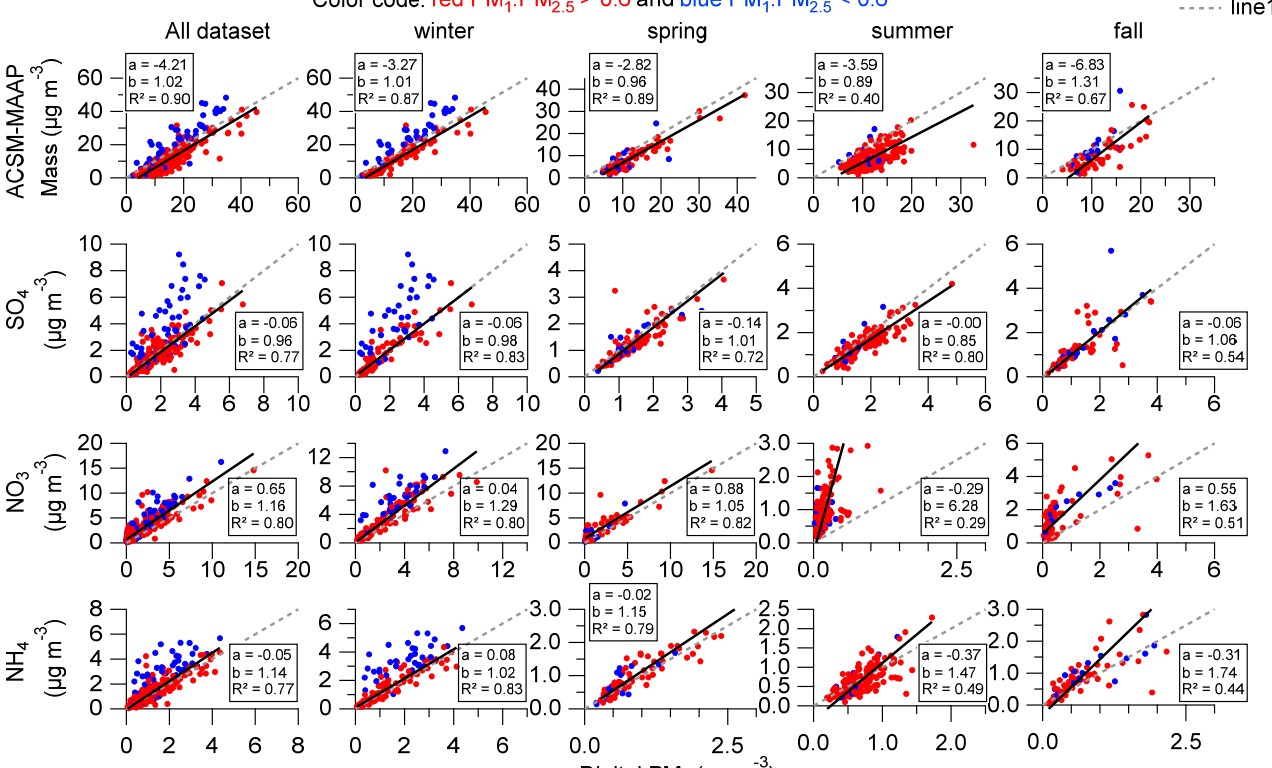

Figure 2: Seasonal variability of the comparison between online and offline PM₁ aerosol measurements. The color coding indicates whether the ratio PM₁:PM₂.₅ total mass concentration is above (red) or below (blue) the selected threshold value of 0.6 (see discussion in section3.1.1.). Dotted grey lines show the line 1:1 and solid black lines represent regression fit by least orthogonal distance fit (y = a + bx). Please note the different axis ranges for the same species.

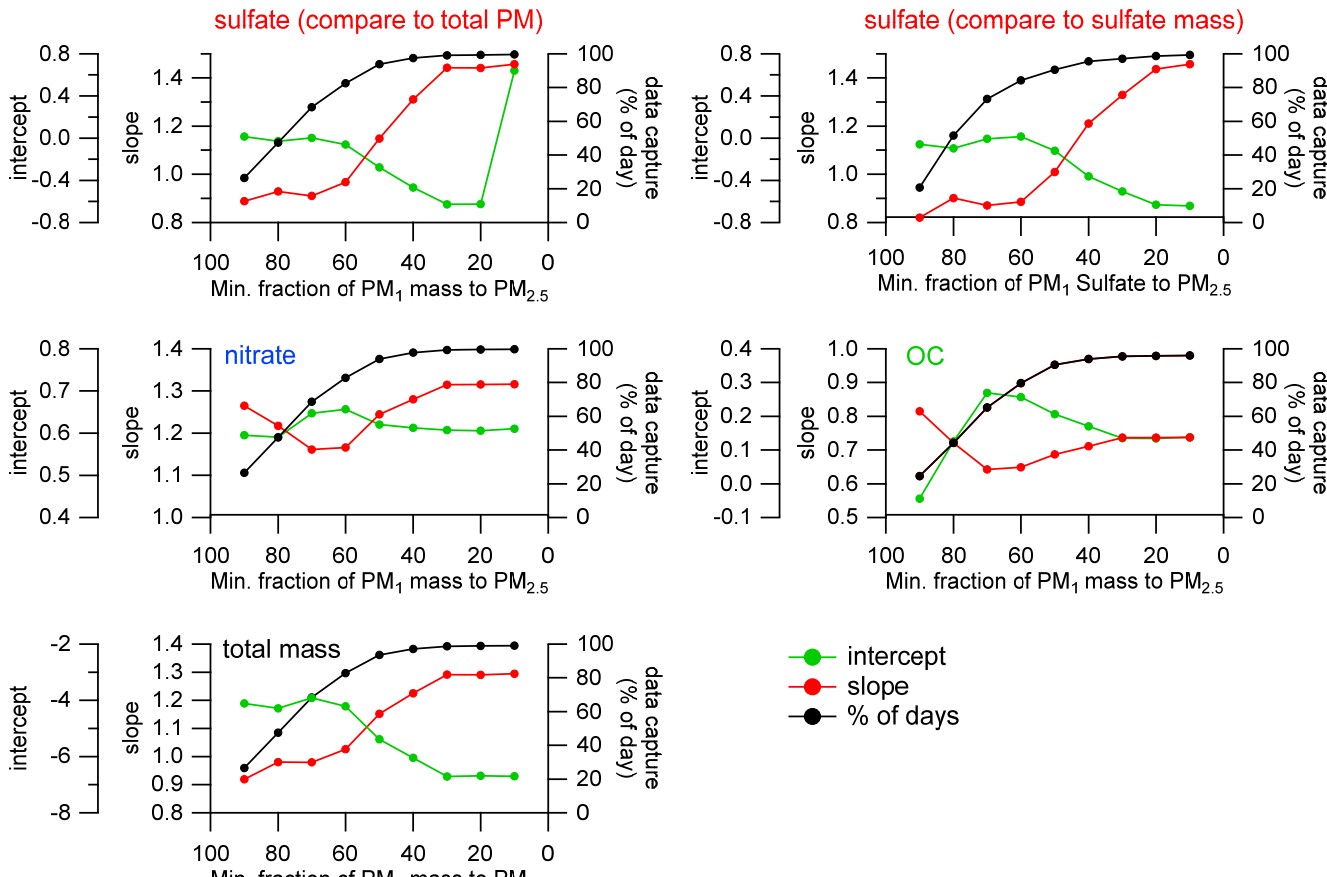

**Figure 3: Sensitivity analysis of the correlation between ACSM and PM$_1$ sulfate, nitrate, OC, and total mass concentration depending on the PM$_1$:PM$_{2.5}$ ratio of the total mass concentration in the range 90 − 10 %. The influence of sulfate distribution on PM$_1$ and PM$_{2.5}$ was also investigated (top left).**

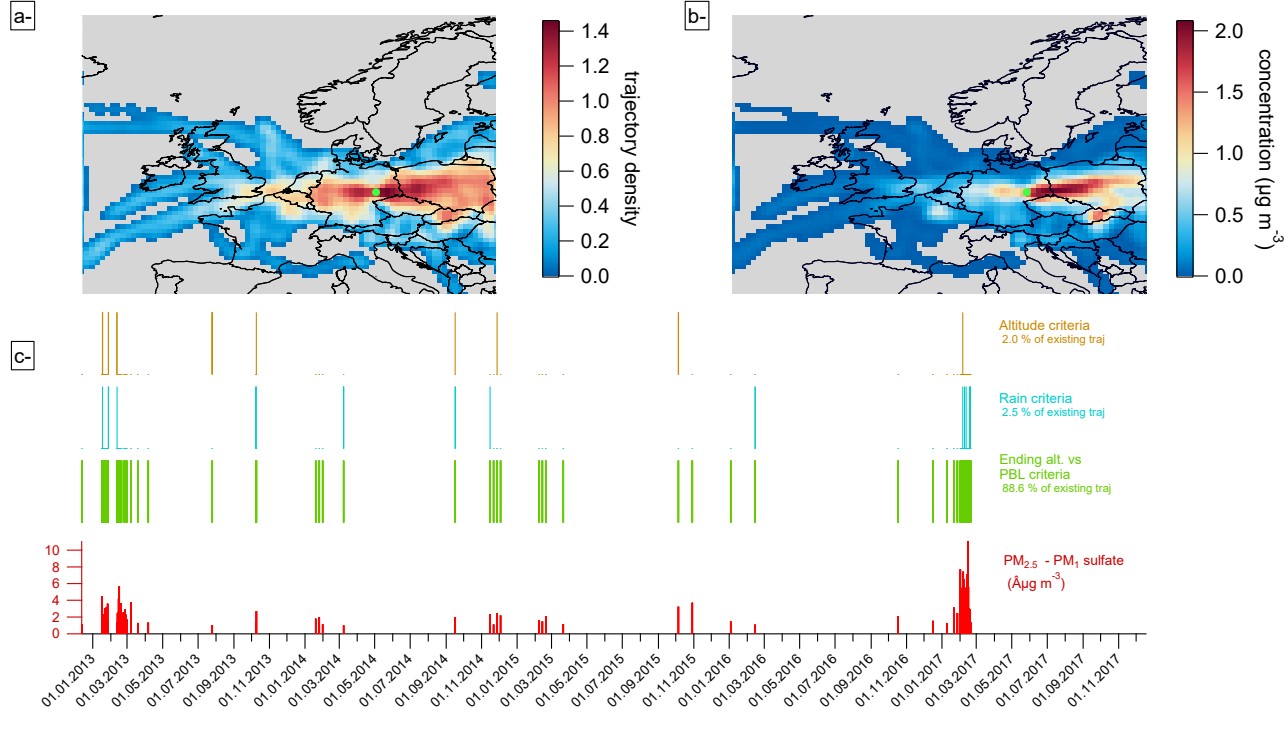

**Figure 4: Trajectory analysis for days where sulfate concentration difference between PM$_{2.5}$ and PM$_1$ exceeded 1 µg m$^{-3}$: (a) overpassing trajectory density; (b) results of the potential source contribution function (PSCF) analysis; (c) time series of sulfate mass concentration difference, trajectory altitude above 2000 m, precipitation events exceeding 1 mm h$^{-1}$ and PBL above the station < 500 m.**

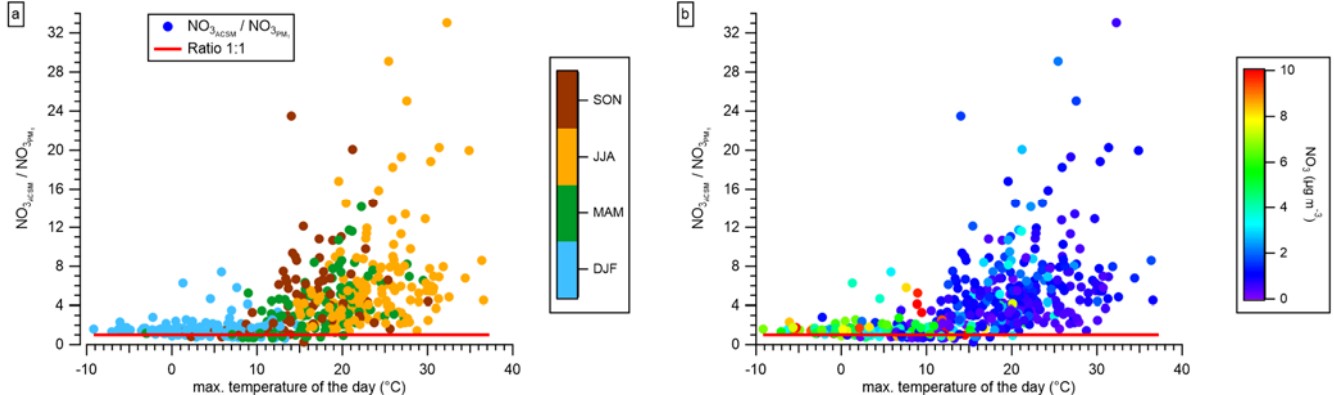

**Figure 5: ACSM:PM₁ ratio nitrate mass concentration compared to the maximum temperature of the corresponding sampling day. The color code corresponds to the different seasons (a) and the total nitrate mass concentration of the ACSM (b).**

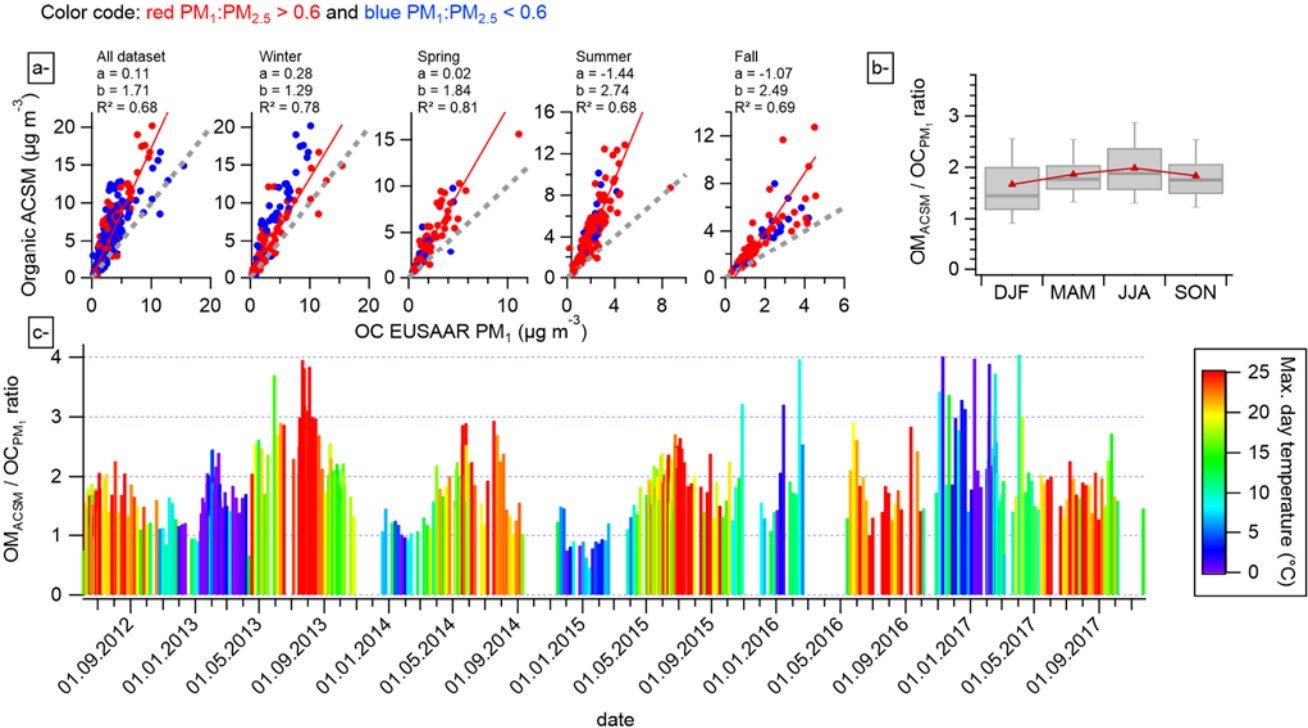

**Figure 6: Correlation between ACSM organic mass concentrations and offline OC PM₁ (a); seasonal variability of the estimated OM$_{ACSM}$:OC$_{PM1}$ ratio (b), and the entire time series colored by maximum daily temperature (c).**

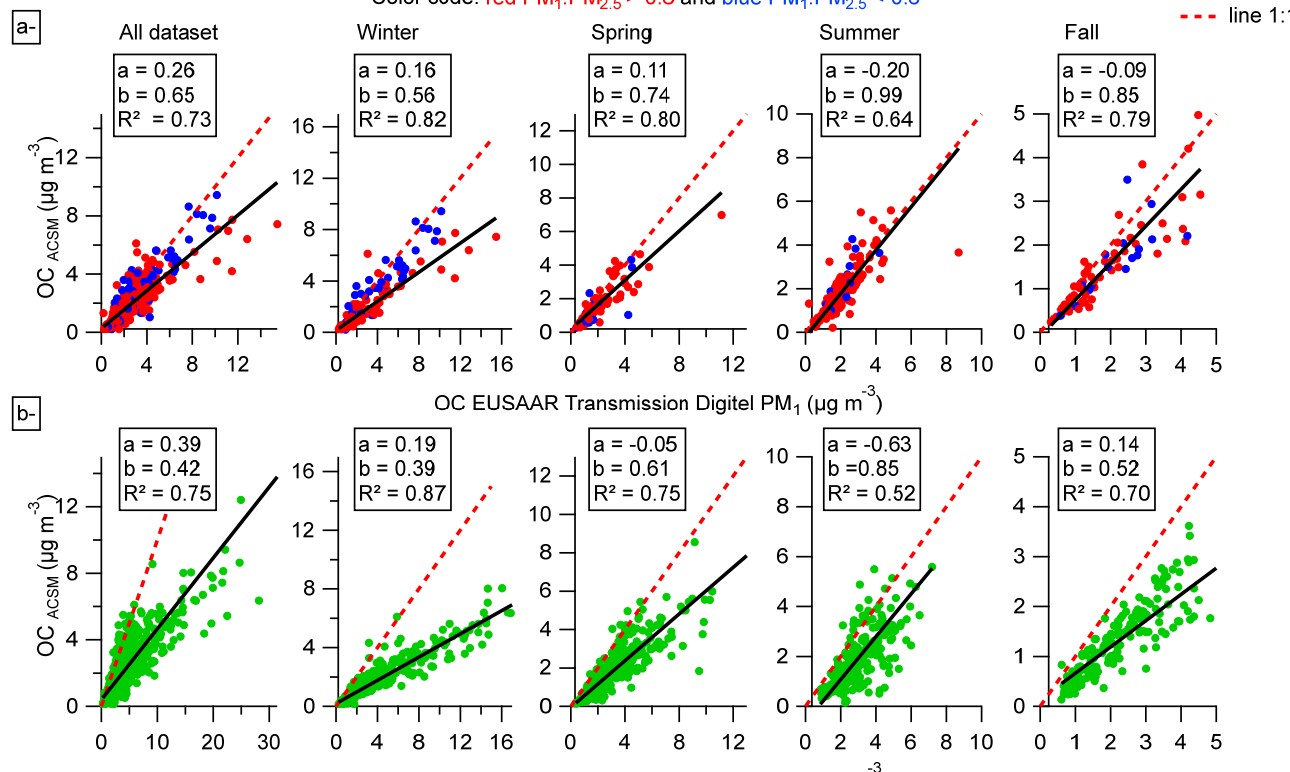

**Figure 7: Correlations between the estimated OC$_{ACSM}$ and the offline OC mass concentration over the entire period and seasonality for PM$_1$ (a), and PM$_{2.5}$ (b). Black lines show the least orthogonal linear fit and the red dotted lines the 1:1 line.**

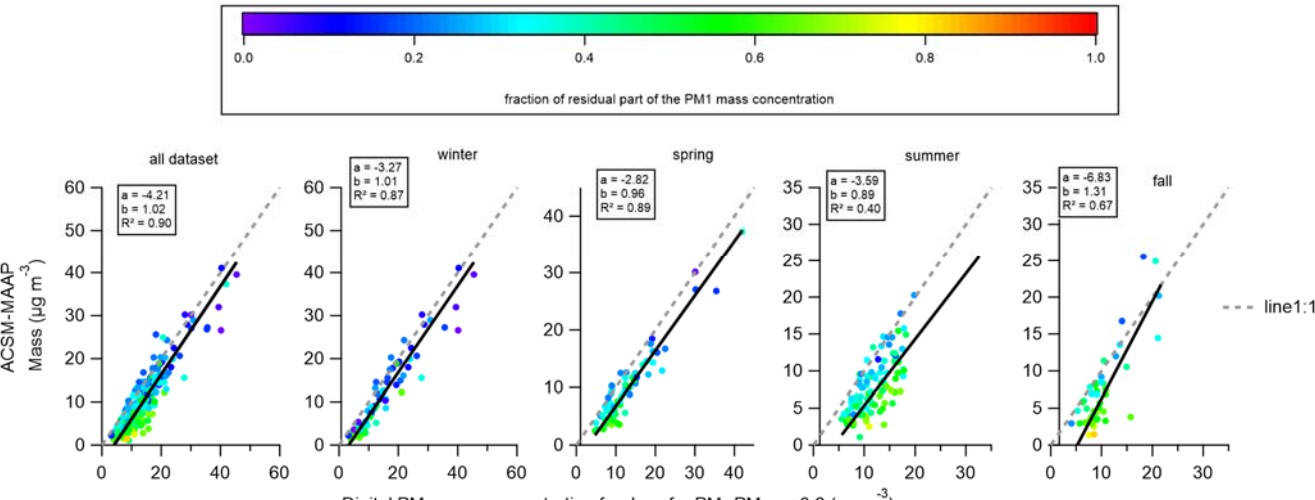

**Figure 8: Influence of the residual mass fraction on the PM₁ filter to the mass closure with online ACSM-MAAP-derived mass concentration.**

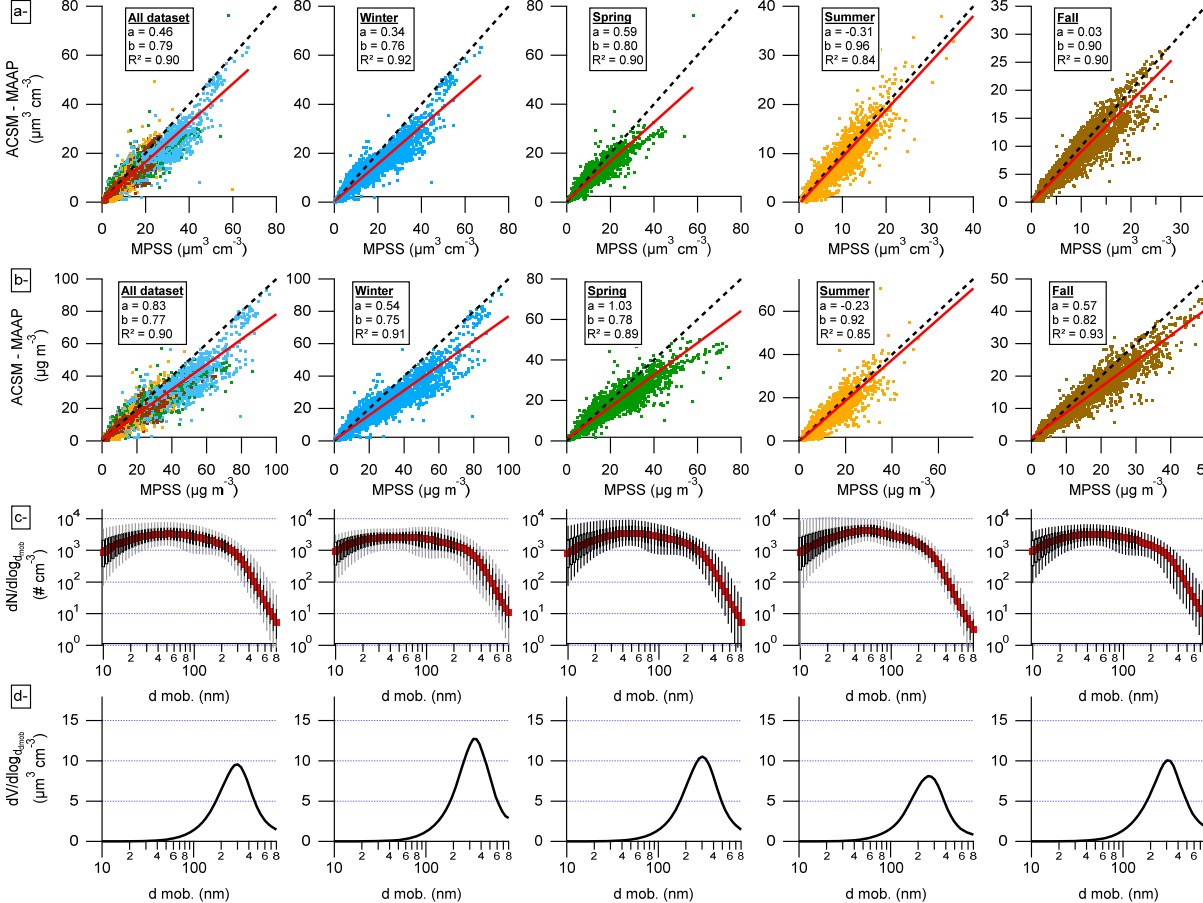

**Figure 9: Comparison between measured ACSM-MAAP and MPSS for the entire period and seasonal variability: volume-closure (a), mass-closure (b), median number size distribution (red) with 10-90 (grey line) and 25-75 (black boxes) percentiles (c), median volume size distribution (d). The linear regressions (red lines) were calculated using the least orthogonal distance fit method.**

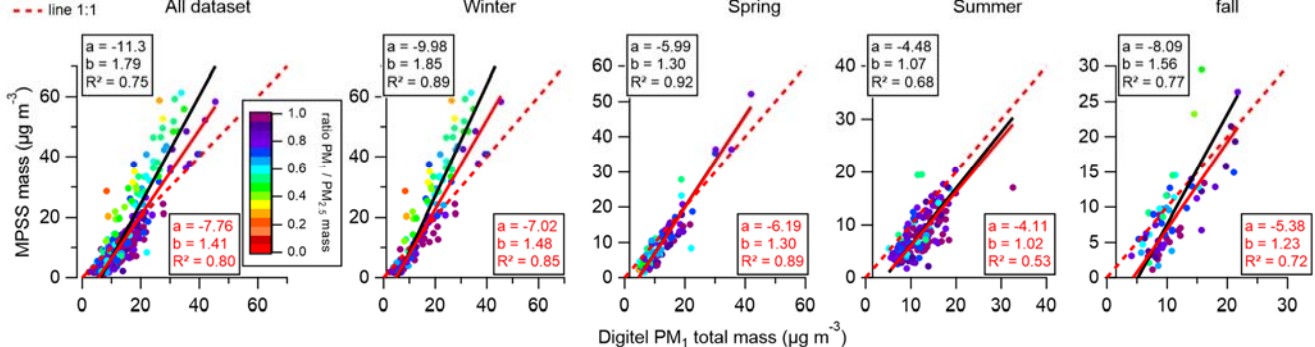

**Figure 10: Comparison between filter PM₁ total mass concentrations and the MPSS-derived mass concentrations for the entire period (left) and the different seasons. The black lines and boxes correspond to the regression fitting without threshold correction and the red lines to the regression fitting according to a PM₁:PM₂.₅ > 0.6.**