# Peer review of "Multi-Year ACSM measurements at the Central European Research Station Melpitz (Germany) Part I: Instrument Robustness, Quality Assurance, and Impact of Upper Size Cut-Off Diameter"

_Atmospheric Measurement Techniques, 2019_

## Referee Comment (RC1) · Anonymous Referee #1 · 12 Oct 2019

The study by Poulain et al. presented a comprehensive evaluation of the ACSM measurements based on the inter-comparisons with other collocated measurements for more than 5 years. The robustness, the limits, and the potential sources of uncertainties of the ACSM measurements for different aerosol species were well discussed. The results are very important for broad ACSM users to understand the long-term measurement uncertainties. The manuscript is well written, and I recommend it for publication.

I have a few comments:

1. The ACSM was calibrated twice at ACMCC, and also could be several times at at the TROPOS research station Melpitz. How robust of the relative ionization efficiencies of ammonium and sulfate were. This is also an important information for long-term measurements.

2. The format of ions should be consistent throughout the manuscript, e.g., page 7, line 15 – 25, use "+" for all ions.

3. Page 15, line 25 – 26, "ACMS" to "ACSM"

---

## Referee Comment (RC2) · Anonymous Referee #2 · 3 Mar 2020

The paper by Poulain et al. reports on long-term ACSM measurements at an established central European location and discuss different aspects of measurement quality assurance. The paper is well developed with data thoroughly analysed and as such is worth publishing in AMT. However, there are several issues to be resolved before it can be accepted for publication. Last but not least English of the paper should be improved following numerous suggestions. Special attention should be given to Figure captions which are often inconsistent and awkwardly worded.

Major comments

Particle mass closure is already a routine requirement requested by reviewers when considering different aspects of AMS/ACSM performance. But what is missing in the abstract and little attention is given in the paper, is the impact of RH and inlets (impactor type or cyclones) affecting comparability. There is no question of the usefulness to cross-checking ACSM with online PM mass or offline PM speciation, but this paper should take a much more comprehensive approach to sampling inlets as crucial factors for maintaining stability and consistency of online chemical speciation monitors, more specifically isokinetic sampling and wet/dry cut-off effects in size selective inlets.

Page 3. Line 26. Volume cannot be converted to mass without the use of average density which is derived from ACSM/AMS measurements making the MPSS derived mass and ACSM-MAAP mass the dependent variables. Instead, ACSM/AMS and MAAP mass can be correctly converted to volume, because individual species mass and density is known (with some exception of organics and black carbon perhaps) making no prior reference to MPSS and keeping both variables independent of each other. All of the above assumes fully internally mixed aerosol which may not always be the case necessitating AMS and BC size distribution.

Page 10. Line 22. I do not follow this reasoning. Sulfuric acid is a stronger acid and ammonia is preferentially neutralizing stronger acid when compared to nitric acid. Therefore, it needs to be checked for degree of neutralization taking into account that ammonium ion has to balance sulfate first and only then the nitrate. Small amounts of organosulfate or organonitrate do not violate the above general pattern at significant sulfate and nitrate concentrations.

Line 30. Estimation of the ratio can only be done after validating ACSM/PM1 agreement which should be done by an independent variable. Otherwise what is the reasoning of using ACSM/PM1or2.5 comparison for sulfate uncovering discrepancies, but assuming that ACSM OM measurements are fine and ready for deriving OM/OC ratio? Later the

authors do comparison by retrieving OC from f44, but that is rather arbitrary given little confidence with regard to AMS measurements done elsewhere and with different instrument. In any event the authors should not start OM comparison going straight to discussing OM:OC ratios before all other technical matters were discussed. And I doubt that ACSM/PM1 comparison can inform about the OM:OC ratio, only AMS/PM1 can, because only AMS can provide OC mass after laborious data processing. Page 11. Line 2. Very high ratio of 2.74 (or even higher observed by Ripoll and Minguillon) suggests that ACSM is probably overestimating OM as well as in the case of sulfate or nitrate with the remainder attributed to OM:OC ratio. Can the authors support such a high oxidation ratio from the literature? I would be very curious to see those studies from the ambient atmosphere as even theoretically the ratio should not exceed ∼3 ($CO_2$ would have 3.66 and it is a gas). It is irrelevant that seasonal OM:OC ratios make sense if they are made from averaging unrealistic values. If so, the authors should not talk about OM:OC ratios before examining slopes carefully and discussing what is and is not theoretically possible. Perhaps consideration of CDCE by neglecting OM is a likely source of overestimation too.

Minor comments

Page 2. Line 7. Spell MPSS for those who read abstract only.

Line 25. . . .composition

Page 3. Line 8. Ovadnevaite et al. have published a rare study that reports...long-term... Do authors suggest in the following sentence that several years (e.g. three) is not sufficiently long period, but their own five year long period is sufficiently long? Please reword or clarify.

Line 18. within the scope, not focus

Line 33. TEOM-FDMS never provides PM1 mass and, even worse, separation in size typically occurs at ambient RH while the mass measurement occurs after drying at

room temperature.

Page 4. Line 9. the impact, not effect.

Line 18. Is the RH actively monitored or indirectly maintained? Is the aerosol flow divided/split isokinetically or randomly? Significant losses can occur if split randomly due to different instrument flow rates.

Page 5. Line 14. ACSM data capture was 80% during deployment at Melpitz.

Line 18. CDCE is Composition Dependent Collection Efficiency as used in the original Middlebrook et al paper. It is also important to note that CDCE algorithm does not take OM into account which can be a potential error source.

Line 23. . . .collect particles with size selective PM2.5 and PM10 inlets on preheated...

Line 24. Samples were collected on a daily basis...

Page 6. Line 15. It is probably meant by "within Planetary Boundary Layer", but I do not quite understand what is meant by "above 500m" and why.

Page 7. Line 2. ...mass concentration comparison is suggesting overestimation by ACSM when compared to offline PM1.

Line 6. influenced by three specific periods in January 2015 and February 2017 (...).

Line 10. The overestimation can be caused by either size selective inlet or(and) sodium sulfate mainly residing in coarse particles. However, I am not sure that is good enough explanation of very contrasting comparison between PM1 and PM2.5. Judging from slopes (1.45 for PM1 and 0.68 for PM2.5) that is suggesting 100% difference between PM1 and PM2.5 - that is massive and not necessarily related to ACSM.

Line 18. That is correct theoretical explanation, but given huge discrepancy during "outlier" days the organic fragment signals should be of very significant magnitude to inorganic ones. Do authors have hints from the multiple campaigns when high resolu-

tion AMS was deployed at the site?

Line 27. coincides, not corresponds.

Line 31. PM1 cannot contribute to PM2.5 it is an inherent part of it. Is it not simply a ratio?

Page 8. Line 1. Consider that wet PM1 size segregation will inevitably pass less submicron particles. However, if that is the plausible culprit then ACSM should perfectly compare with PM2.5 which even in wet cut-off conditions should well correspond to dry ACSM PM1.

Line 5. The effect may still be there, but masked by the evaporation effect and NaNO3 effect on filters. Is this sentence out of place as it is becomes confusing whether the following sentences regard to sulfate or nitrate?

Line 19. Please explain what air mass density means.

Line 20. This confirms the predominantly shallow PBL...

Line 28. ...to slightly overestimate... The overestimation is very small and within the error margin compared to massive overestimation in Minguillon et al. I see this as a contrast from previous studies, not similarity.

Page 9. Line 5. I see more evidence in Figure 5. First, a lot of nitrate evaporation is visible at low ambient temperature as many of the points are clearly above ratio of 1.0. That is not surprising considering evaporation from particles already collected on the filter. However, the effect in summer seems to be larger, but concentrations are much lower in summer suggesting that evaporation can be near constant (as long as filter temperature is maintained similar during all seasons which needs to be confirmed). As more nitrate loaded particles are deposited on filter during winter they become buried under new layers of particles before being significantly evaporated. During summer there is little nitrate on particles and less particles altogether resulting in more absolute evaporative losses until particles buried under the new layers of particles.

Page 10. Line 20. ...with off-line measurements fall somewhere in between the two previously discussed ions.

Page 12. Line 10. in another location.

Line 25. The first comparison is correct by comparing mass with mass, but the second comparison is not, because particle mass derived from PNSD and MPSS will rely on ACSM for estimating average density making the two variables dependent. Therefore, only volume, not mass comparison of ACSM-MAAP and PNSD can be done correctly.

Page 13. Line 3. Therefore, it is not possible to conclude whether....depends on the location or the presence of larger coarse mode.

Line 9. ...cannot typically account for the entire mass

Line 12. It is inappropriate to arbitrarily choose specific OM:OC ratio when the above paragraphs discussed wildly different ratios.

Line 32. Incorrect method. See above.

Page 14. Line 24. This artefact may plausibly explain the seasonality of the mass closure (it should be volume closure instead).

Page 15. Line 1. By the same principle ACSM size range is 1um/1.6=0.625um (dvac/density=dmob) and discrepancies between size ranges of ACSM and MPSS should be irrelevant. Please reconsider and reflect in conclusions.

Line 31. Can this be called an artefact? It depends whether organo-nitrate belongs to organics class or nitrate. Nitrate functional group is still a nitrate even if bound to organic species. Quite contrary, off-line inorganic and offline OC/EC analysis completely misses nitrate bound to organics making the use of the term "artefact" justified. Ideally, one would want a clear distinction of organo-nitrate compound which would be out of reach by off-line inorganic techniques unless specifically measured for OrgNO3.

Figure 1. I recommend changing PM to ACSM-MAAP or ACSM+eBC for consistency

with further Figures.

Figure 2. ...and solid black lines represent regression fit by least orthogonal distance (y=a+bx)

Figure 3. change "data coverage" to "data capture".

Figure 4. ...for days where sulfate concentration difference between PM2.5 and PM1 exceeded 1ug/m3: (a) overpassing trajectory density; (b) potential source contribution function??; (c) time series of sulfate concentration difference, PBL height above 2000m, precipitation events exceeding 1mm/h and PBL formation above the station altitude.

Figure 6. ...and off-line OC PM1 (a); seasonal variability....(b) and the entire time series (c).

Figure 7. Awkward Figure caption. Please rewrite according to suggested above.

Figure 9. Incorrect graph as it should be volume comparison. Awkward Figure caption. Please rewrite according to suggested above.

Figure S1. Make PM axis consistent with further Figures

Figure S2. ...to daily PM2.5 mass.

Figure S5. ...corrected for organo-nitrate contribution during days with PM1:PM2.5 < 0.6.

---

## Author Response (AR1)

The study by Poulain et al. presented a comprehensive evaluation of the ACSM measurements based on the inter-comparisons with other collocated measurements for more than 5 years. The robustness, the limits, and the potential sources of uncertainties of the ACSM measurements for different aerosol species were well discussed. The results are very important for broad ACSM users to understand the long-term measurement uncertainties. The manuscript is well written, and I recommend it for publication.

We would like to thank the referee for his/her constructive comments and suggestions made to improve and clarify our manuscript. Our responses are given below. For clarity, comments from the referee are in black, our responses in blue, and change on the text of the manuscript in **bold blue**.

The authors would like to mention here that two additional co-authors were added to the initial list regarding to their contributions on the revision of the manuscript.

I have a few comments:

R1. The ACSM was calibrated twice at ACMCC, and also could be several times at the TROPOS research station Melpitz. How robust of the relative ionization efficiencies of ammonium and sulfate were. This is also an important information for long-term measurements.

A1: Thanks for bringing up this interesting and important point. To consider this, the following text has been added to the ACSM description (section 2.2) that discussed calibration and stability of the ionization efficiency (IE) and relative ionization efficiency (RIE) of ammonium and sulfate:

**"The ACSM was regularly calibrated according to the manufacturer's recommendations at that time with 350 nm monodispersed ammonium nitrate and ammonium sulfate particles selected by a DMA and using the jump scan approach. It is important to note that since recently, the recommended calibration method has changed to a full scan approach (Freney et al., 2019). The total particle number concentration was systematically set below 800 # cm$^{-3}$ to limit the artefact due to multiple charged particles. An overview of the ionization efficiency (IE) and relative ionization efficiency (RIE) for ammonium and sulfate can be found in Figure SI-1. On average, all performed calibration provides a mean IE value of 4.93 (± 1.45) 10$^{-11}$ (mean ± std. dev.) and mean RIEs for ammonium and sulfate were 6.48 ± 1.26 and 0.68 ± 0.13, respectively. These values are very close to the ones used for the data evaluation as indicated in Figure SI-1. Overall, no clear trend for IE and RIE of sulfate can be observed over the period, while a small decrease in the RIE of ammonium can be reported. The lowest RIE of ammonium was reported just after the replacement of the filament indicated a possible need for degassing and stabilization period. However, it is difficult to conclude if these tendencies could be associated with a possible aging effect of the instrument since it corresponds to a single instrument. Similar observations on various other individual ACSMs would be needed to allow for stating such a conclusion and a more systematic investigation of potential trends should then be performed with a large number of ACSM."**

[Figure]

**Figure SI-1:** Time variation of the IE and RIE for ammonium and sulfate. The single points correspond to calibration, the dashed black line to the mean value from the calibration, and the full red line the mean value from the data analysis (shaded area corresponds to the standard deviation). Major maintenance (change of filament and vaporizer) are including.

R2. The format of ions should be consistent throughout the manuscript, e.g., page 7, line 15 – 25, use "+" for all ions.

A2: Charge was added to all mass spectra fragments.

Following correction were made:

P7, L 15-27: "[…] same m/z (for example, $C_6H_8^+$ and/or $C_5H_4O^+$ at m/z 80 for $SO_3^+$, or $C_6H_9^+$ and $C_5H_5O^+$ at m/z 81 for $HSO_3^+$) […]  change of $SO_3^+/SO^+$ and $HSO_3^+/SO^+$ […]"

P 9, L 11: "[…] at m/z 30 ($NO^+$) and m/z 46 ($NO_2^+$), as well as on a minor contribution of $N^+$ and $HNO_3^+$ ions […] "

P11, l31: "[…] a possible artefact on the $CO_2^+$ signal itself."

R3. Page 15, line 25 – 26, "ACMS" to "ACSM"

A3 Corrected

**References**

Freney, E., Zhang, Y., Croteau, P., Amodeo, T., Williams, L., Truong, F., Petit, J.-E., Sciare, J., Sarda-Esteve, R., Bonnaire, N., Arumae, T., Aurela, M., Bougiatioti, A., Mihalopoulos, N., Coz, E., Artinano, B., Crenn, V., Elste, T., Heikkinen, L., Poulain, L., Wiedensohler, A., Herrmann, H., Priestman, M., Alastuey, A., Stavroulas, I., Tobler, A., Vasilescu, J., Zanca, N., Canagaratna, M., Carbone, C., Flentje, H., Green, D., Maasikmets, M., Marmureanu, L., Minguillon, M. C., Prevot, A. S. H., Gros, V., Jayne, J., and Favez, O.: The second ACTRIS inter-comparison (2016) for Aerosol Chemical Speciation Monitors (ACSM): Calibration protocols and instrument performance evaluations, Aerosol Sci. Technol., 1-25, 10.1080/02786826.2019.1608901, 2019.

The paper by Poulain et al. reports on long-term ACSM measurements at an established central European location and discuss different aspects of measurement quality assurance. The paper is well developed with data thoroughly analysed and as such is worth publishing in AMT. However, there are several issues to be resolved before it can be accepted for publication. Last but not least English of the paper should be improved following numerous suggestions. Special attention should be given to Figure captions which are often inconsistent and awkwardly worded.

We would like to thank the referee for his/her constructive comments and suggestions made to improve and clarify our manuscript. Our responses are given below. For clarity, comments from the referee are in black, our responses in blue, and change on the text of the manuscript in **bold blue**.

The authors would like to mention here that two additional co-authors were added to the initial list regarding to their contributions on the revision of the manuscript.

**Major comments**

R.1: Particle mass closure is already a routine requirement requested by reviewers when considering different aspects of AMS/ACSM performance. But what is missing in the abstract and little attention is given in the paper, is the impact of RH and inlets (impactor type or cyclones) affecting comparability. There is no question of the usefulness to cross-checking ACSM with online PM mass or offline PM speciation, but this paper should take a much more comprehensive approach to sampling inlets as crucial factors for maintaining stability and consistency of online chemical speciation monitors, more specifically isokinetic sampling and wet/dry cut-off effects in size selective inlets.

A1: The reviewer points out several important questions that will be answered separately.

1. The impact of RH when comparing measurements made at ambient RH (typically offline samplers) with the ones made at dry conditions.

We agree that mentioning this aspect is missing in the manuscript. However, we would like to draw the attention of the referee that our group already published a paper specifically discussing this question: Chen, Y., Wild, O., Wang, Y., Ran, L., Teich, M., Gross, J., Wang, L. N., Spindler, G., Herrmann, H., van Pinxteren, D., McFiggans, G., and Wiedensohler, A.: The influence of impactor size cut-off shift caused by hygroscopic growth on particulate matter loading and composition measurements, Atmos. Environ., 195, 141-148, 2018. Based on this study, the cut-off shift due to aerosol hygroscopic growth should play a minor role at Melpitz, as this effect was estimated to influence the comparison by 2 % for marine air-mass and 1 % for continental air-mass for $PM_1$ (7.2 % and 1.1 %, respectively for the $PM_{2.5}$). Overall for the European background station, the cut-off shift represents less than 10 % for $PM_1$ and 20 % for $PM_{2.5}$ particle mass loading. However, the cut-off shift can be stronger for marine or coastal stations (up to 43 % for $PM_1$ and 62 % for $PM_{2.5}$) and must consequently be considered when doing such a comparison.

The following sentences have been added at the beginning of section 3.1 before starting discussing the comparisons between ACSM and offline samplers:

"**It is also important to note here that the comparison between ACSM and offline samplers generally consists of comparing dry aerosol online measurements to offline analyses of samples collected at ambient RH. A direct consequence is that the offline results might suffer from a cut-off shift due to aerosol hygroscopic growth when ambient RH is high (Chen et al., 2018). Based on**

**this study, the cut-off shift due to aerosol hygroscopic growth should play a minor role at Melpitz, as this effect was estimated to influence the comparison by 2 % for marine air-mass and 1 % for continental air-mass. For European background stations, such a cut-off shift has been estimated to represent less than 10 % for PM$_1$ and 20 % for PM$_{2.5}$ particle mass loading, while it is stronger for marine or coastal stations (up to 43 % for PM$_1$ and 62 % for PM$_{2.5}$). Therefore, such artefact has to be considered when comparison ACSM with offline measurements."**

The following sentence has been added to the conclusion:

"Therefore, for such a comparison, the limitations due to the different size cuttings must be considered. **Moreover, possible cut-off shift due to ambient relative humidity effect on the offline measurements could represent a non-negligible parameter and has to be considered during such an exercise, especially for marine stations.**"

2. Inlet type (impactor or cyclone): we agree with the referee on the importance of the type of the inlet for aerosol measurements and later comparison. However, this effect was not investigated in this study since only one inlet was used. Nevertheless, inlet system for offline filter-based analyses are defined on the Air Quality Directive EN 12341 of the European Union. For the online instruments, the WMO/GAW recommend to use either a cyclone or an impactor with an upper cut point of 10 μm (WMO/GAW, 2016). The use of such a high cut point inlet makes the influence of the inlet type for sub-μm particle negligible.

3.Isokinetic sampling:

Because all the instruments connected to the main sampling line do not have the same flow rate, it is mandatory to ensure that the splitting sampling flow between all of them is made in a representative way. This is the reason why the sampling flow distribution at Melpitz is made by an isokinetic splitter (Fig. 1) following the GAW and ACTRIS recommendation (https://www.wmo-gaw-wcc-aerosol-physics.org/files/actris-recommendation-for-aerosol-inlets-and-sampling-tubes.pdf).

[Figure]

Figure 1: Example sketch of the isokinetic splitter similar to the one used on the sampling line at Melpitz (copy from https://www.wmo-gaw-wcc-aerosol-physics.org/files/actris-recommendation-for-aerosol-inlets-and-sampling-tubes.pdf).

We have corrected the text (section 2.1) to mention the isokinetic splitter on the station description as follows:

"the aerosol flow is divided among a set of instruments" was changed to "**The aerosol flow is divided among a set of instruments by an isokinetic splitter (WMO/GAW, 2016) ensuring a representative sampling between the instruments**."

In conclusion, the following sentence was added: "**For such an exercise, it is fundamental to ensure isokinetic flow splitting between the different instruments connected to the main sampling line to ensure a homogeneous distribution of the air sample**."

Finally, the following changes have also been made on the abstract and the introduction:

"**The Aerosol Chemical Speciation Monitor (ACSM) is nowadays widely used to identify and quantify the main component of fine particles in ambient air. As such, its deployment at observatory platforms is fully incorporated within the European Aerosol, Clouds and Trace Gases Research Infrastructure (ACTRIS). To ensure the consistency** of the dataset, as well as instrumental performance and variability, regular intercomparisons are organized at the Aerosol Chemical Monitoring Calibration Center (ACMCC, part of the European Center for Aerosol Calibration, Paris, France). However, in-situ quality assurance remains a fundamental tracking point of the instrument's stability. **Here, we present and discuss the main outputs of long-term quality assurance efforts achieved for ACSM measurements at the research station Melpitz (Germany) since 2012 onwards. In order to validate the ACSM measurements over the years and to characterize the seasonal variations, nitrate, sulfate, ammonium, organic, and particle mass concentrations […]**"

"For this purpose, a European **distributed facility of ground-based Aerosol Chemical Species Monitor (ACSM, Ng et al., 2011) is operated within ACTRIS (European Research Infrastructure for the observation of Aerosol, Clouds and Trace Gases, http://www.actris.eu). Complementary, the COST Action CA16109 Chemical On-Line cOmpoSition and Source Apportionment of fine aerosol (COLOSSAL, https://www.costcolossal.eu) is gathering a wide community of European research groups (with even further international inputs, as well as participation of some regional air quality monitoring networks) interested in the fine aerosol fraction. One of the main objectives of these coordinated programs is to investigate and understand the spatial variability […]**"

R2: Page 3. Line 26. Volume cannot be converted to mass without the use of average density which is derived from ACSM/AMS measurements making the MPSS derived mass and ACSM-MAAP mass the dependent variables. Instead, ACSM/AMS and MAAP mass can be correctly converted to volume, because individual species mass and density is known (with some exception of organics and black carbon perhaps) making no prior reference to MPSS and keeping both variables independent of each other. All of the above assumes fully internally mixed aerosol which may not always be the case necessitating AMS and BC size distribution.

A2: It is true that using the density derived from the chemical composition of the ACSM-MAAP to determine the MPSS-derived mass concentration makes the two measurements dependent on each other. However, atmospheric aerosol measurements are generally performed in mass concentration making the quantification of a potential discrepancy more apprehensible in terms of mass ($\mu g\ m^{-3}$) than in volume ($\mu m^3\ cm^{-3}$). This may be the reason why the mass approach is more common in the literature than the volume one. Nevertheless, and as suggested by the referee, we reinvestigated all our comparisons using

volume concentration and assuming spherical particles, fully internally mixed and identical chemical composition over the entire size distribution. As shown in Figure 2, changing the unit does not influence the slopes between the two instruments and did not affect our conclusions. As a consequence, regarding our sampling place, the two approaches appear to be similar.

[Figure]

**Figure 2: Comparison between measured ACSM-MAAP and MPSS for the entire period and seasonal variability: volume-closure (a), mass-closure (b), median number size distribution (red) with 10-90 (grey line) and 25-75 (black boxes) percentiles (c), median volume size distribution (d). The correlation curves (red line) were calculated using the least orthogonal distance fit method.**

The text at the beginning of section 3.2.2. has been changed as follow:

**"The PNSD has been continuously measured in parallel to the aerosol mass spectrometer and can, therefore, be used to perform mass closure analysis between ACSM-MAAP and PNSD (ranging from 10 to 800 nm, mobility diameter). To ensure a robust comparison between the two systems, two approaches are reported in the literature: the first one consists of converting the ACSM-MAAP mass concentration into volume and the PNSD in volume concentration. The second one consists of converting the PNSD into mass concentration. Both approaches are based on the same assumptions of (i) spherical, (ii) fully internally mixed particles, and (iii) an identical chemical composition over the entire size distribution to estimate a chemical time-dependent gravimetric particle density based on the following equation from Salcedo et al. (2006):**

$$density = \frac{[Total_{AMS} + eBC]}{\frac{[NO_3^-] + [SO_4^{2-}] + [NH_4^-]}{1.75} + \frac{[Cl^-]}{1.52} + \frac{[Org]}{1.2} + \frac{[eBC]}{1.77}}$$  (1)

Here, the density was assumed to be 1.75 g cm⁻³ for ammonium nitrate and ammonium sulfate (Lide, 1991), 1.52 g cm⁻³ for ammonium chloride (Lide, 1991), and 1.2 g cm⁻³ for organic matter (Turpin and Lim, 2001). Finally, a density of 1.77 g cm⁻³ (Park et al., 2004) was applied for eBC. A discussion of eBC density can be found in Poulain et al. (2014).

It is important to note that for the volume concentration approach, both measurements (ACSM-MAAP and MPSS) remain independent between each other, which is not the case when using mass concentration. However, only a few numbers of papers reported a comparison between AMS or ACSM and MPSS in volume concentration (e.g. DeCarlo et al., 2008;Elsasser et al., 2012). Even though the two variables are non-independent on the mass concentration approach, it remains the most commonly used. A possible reason is that the mass concentration unit remains easier to use and interpreted as the volume concentration since atmospheric measurements are usually made in mass concentration.

Here, we investigated comparison results obtained using each of these approaches. Results are summarized in Figure 9.a for the volume concentration approach and Figure 9.b for the mass concentration one.

[Figure]

**Figure 9: Comparison between measured ACSM-MAAP and MPSS for the entire period and seasonal variability: volume-closure (a), mass-closure (b), median number size distribution (red) with 10-90 (grey line) and 25-75 (black boxes) percentiles (c), median volume size distribution (d). The linear regressions (red lines) were calculated using the least orthogonal distance fit method.**

Over 5.5 years of measurements, the ACSM-MAAP-derived volume or mass concentration correlates well with the estimated volume or mass concentration of the MPSS with the same slope of 0.79 (R² = 0.90, Fig. 9a and 9b). This matches similar previous comparisons at the same place with

an AMS (Poulain et al., 2014). **Therefore, the selected method (volume or mass) did not substantially influence the comparison results. In the conditions of the present study both approaches could be applied for the station of Melpitz. Since comparison in mass concentration is the more commonly used, we will focus on it in the following discussions."**

In summary, the following comment was added:

**"It can be performed by converting the ACSM-MAAP mass concentration into volume concentration or by converted the MPSS volume concentration into mass concentration both using time-dependent density and assuming spherical and fully internally mixed particles. The volume approach is the most robust since it enables a strictly independent method. Being more interpretable, the mass approach may be used instead of for error quantification as long as it agrees with the volume approach. For the present dataset, the selected method did not substantially influence the comparison results."**

R3: Page 10. Line 22. I do not follow this reasoning. Sulfuric acid is a stronger acid and ammonia is preferentially neutralizing stronger acid when compared to nitric acid. Therefore, it needs to be checked for degree of neutralization taking into account that ammonium ion has to balance sulfate first and only then the nitrate. Small amounts of organosulfate or organonitrate do not violate the above general pattern at significant sulfate and nitrate concentrations.

A3: As suggested by the referee, a discussion on the neutralization of ammonium by nitrate, sulfate, and chloride species for both ACSM and $PM_1$ filters was added in section 3.1.3 before discussing the comparison of the two methods.

The section has been rewritten and a Figure has been added to the Supplementary information as follow:

**"The ammonium mass concentration measured by the ACSM mostly corresponds to ammonium nitrate and ammonium sulfate salts. Before comparing ACSM and offline $PM_1$ ammonium mass concentration, the neutralization state of the particles was estimated for both datasets assuming a full neutralization by nitrate, sulfate, and chloride as described in e.g. Sun et al. (2010). In both approaches, particles can be considered as fully neutralized during the entire period with no seasonality (Fig. SI-7) in agreement with previous AMS measurements made at the same place (Poulain et al., 2011). Correlations with offline systems fall somewhere between the two previously discussed ions. During the cold season, the ACSM ammonium mass concentration matches the $PM_1$ (slope 1.02, $R^2 = 0.83$), which supports the larger fraction of ammonium nitrate in the total PM as well as the size effect of sulfate during wintertime (Fig. 1 and Fig. 2). During the warm season, the evaporation of ammonium nitrate as discussed before will also induce a loss of ammonium on the filter samples compared to the online measurements leading to an under-estimation of the ammonium concentration on the offline sampler as well as a poor correlation ($R^2 = 0.49$). Similar conclusions can also be drawn when comparing it to the $PM_{2.5}$ ammonium mass concentration Fig. SI-1 & SI-2)."**

[Figure]

**Figure SI-5: Comparison of the measured ammonium with the predicted ammonium mass concentration for the ACSM (left) and offline PM₁ samples assuming a full neutralization by nitrate, sulfate, and chloride. The linear regressions were calculated using the least orthogonal distance fit method (y = a x + b).**

R4: Line 30. Estimation of the ratio can only be done after validating ACSM/PM1 agreement which should be done by an independent variable. Otherwise what is the reasoning of using ACSM/PM1or2.5 comparison for sulfate uncovering discrepancies, but assuming that ACSM OM measurements are fine and ready for deriving OM/OC ratio? Later the authors do comparison by retrieving OC from f44, but that is rather arbitrary given little confidence with regard to AMS measurements done elsewhere and with different instrument. In any event the authors should not start OM comparison going straight to discussing OM:OC ratios before all other technical matters were discussed. And I doubt that ACSM/PM1 comparison can inform about the OM:OC ratio, only AMS/PM1 can, because only AMS can provide OC mass after laborious data processing.

A4: We agree with the first part of this comment. A direct comparison of the organic mass concentration measured by the ACSM with collocated organic mass concentration would be the ideal case and would allow us to proceed on a comparable approach as for the inorganic species. However, this approach is not possible, since no other instrument has provided direct OM measurements at the station. Consequently, the best way to directly compare organic measurements together is to either use the OM:OC ratio or to compare OC values. Nevertheless, we agree that limitations on both measurements must be first discussed further. For this purpose, Figure 6c of the manuscript was modified to consider temperature effect on filter samples. Consequently, the text of the manuscript was modified and reorganized following the reviewer's suggestions.

We would like to underline here that using the contribution of the $CO_2^+$ signal to the total organic signal ($f_{44}$) as a surrogate to estimate the OM:OC ratio is commonly used within the AMS community when working on Unit Mass Resolution organic mass spectra (e.g. Chen et al., 2015 and reference therein). A comparison with the elemental analysis approach using high-resolution organic mass spectra of the HR-ToF-AMS was successfully made by Aiken et al. (2008) and Canagaratna et al. (2015). The latter one concluded on the accuracy of 13 % of the $f_{44}$ method compared to the elemental analysis one for the SOA compounds, while this accuracy is decreasing for primary OA standards having an $f_{44}$ < 4% on average. Considering that m/z 44 dominates the organic mass spectra, it is reasonable to consider the 13 % accuracy as the highest accuracy that can be obtained for unit mass resolution AMS results. Consequently considering, that m/z 44 is systematically the dominant fragment of the organic mass spectra for ambient measurements and that the ACSM is based on similar principle than the AMS, it is still relevant to apply the $f_{44}$ approach on the ACSM organic results as a proxy for ambient OC, and compare the results with the well-established offline OC method. Finally, on a longer perspective, presenting and discussing results from such a comparison whenever possible is also important in the frame of the standardization process, currently trying to establish guidelines for investigating possible equivalence towards the standard EC-OC offline method of any type of alternative measurement technique.

As a result, the text on section 3.1.4 was changed as follows:

[revised manuscript text omitted]

R5: Page 11. Line 2. Very high ratio of 2.74 (or even higher observed by Ripoll and Minguillon) suggests that ACSM is probably overestimating OM as well as in the case of sulfate or nitrate with the remainder attributed to OM:OC ratio. Can the authors support such a high oxidation ratio from the literature? I would be very curious to see those studies from the ambient atmosphere as even theoretically the ratio should not exceed ~3 ($CO_2$ would have 3.66 and it is a gas). It is irrelevant that seasonal OM:OC ratios make sense if they are made from averaging unrealistic values. If so, the authors should not talk about OM:OC ratios before examining slopes carefully and discussing what is and is not theoretically possible. Perhaps consideration of CDCE by neglecting OM is a likely source of overestimation too.

A5: We agree with the referee and invite the referee to read our answer A4, which also includes a discussion on the uncertainties of the two approaches (ACSM and offline).

Minor comments

R6: Page 2. Line 7. Spell MPSS for those who read abstract only.

A6: added "[…] of the Mobility Particle Size Spectrometer (MPSS) […]"

R5: Line 25. . . . .composition

A7: corrected

R8: Page 3. Line 8. Ovadnevaite et al. have published a rare study that reports...longterm... Do authors suggest in the following sentence that several years (e.g. three) is not sufficiently long period, but their own five years long period is sufficiently long? Please reword or clarify.

A8: Three years is indeed long enough to check the stability of an instrument. However, the authors compared their near-$PM_1$ HR-ToF-AMS data with $PM_{2.5}$ samples and did not discuss the variability of the correlation over these 3-years. Therefore, we considered that our approach provides more understanding of such a systematic comparison to identify sources of deviation and artefacts.

We have clarified it as follows: "Ovadnevaite et al. (2014) have written a rare published work that reports long-term AMS comparisons **(3-years). Even though the authors successfully compared it with offline $PM_{2.5}$ filter samples, they did not discuss the variability of the correlation and potential source of uncertainties.**"

R9: Line 18. within the scope, not focus

A9: Changed

R10: Line 33. TEOM-FDMS never provides PM1 mass and, even worse, separation in size typically occurs at ambient RH while the mass measurement occurs after drying at room temperature.

A10: We are not certain to fully understand this comment. The size cutting of a TEOM-FDMS depends on the aerosol inlet used. When connecting to $PM_1$ inlet, the TEOM-FDMS will provide $PM_1$ aerosol mass concentration. Moreover, it is true that the inlet cut-off occurs at ambient RH, while the mass measurements after drying at room temperature, which is a general issue for all online aerosol measurements. The use of an impactor after conditioning the sampling air, while being certainly more efficient for keeping a constant cutoff on the measurement.

Please refer to our answer A1 for the discussion on the effect of RH on aerosol sampling. Nevertheless, discussing the interaction on the RH to the TEOM-FDMS measurements made during the cited study is behind the scope of the present manuscript.

R11: Page 4. Line 9. the impact, not effect.

A11: Changed

R12: Line 18. Is the RH actively monitored or indirectly maintained? Is the aerosol flow divided/split isokinetically or randomly? Significant losses can occur if split randomly due to different instrument flow rates.

A12: RH is actively monitored on the sampling line by an automatic aerosol diffusion dryer to keep the relative humidity on the sampling line below 40 %. A detailed description of the drying system can be found on the cited reference of Tuch et al., 2009.

The text has been changed to include the word "actively":

"This inlet line consists of a $PM_{10}$ Anderson impactor located approximately 6 m above ground level and directly followed by an automatic aerosol diffusion dryer to **actively** keep the relative humidity on the sampling line below 40 % (Tuch et al., 2009)."

As mentioned in our answer to the first major comment (A1), an isokinetic splitter (Fig. 1) is used to ensure a representative sampling between all the connected instruments.

The text has been changed as follows:

"**The aerosol flow is divided among a set of instruments by an isokinetic splitter (WMO/GAW, 2016) ensuring a representative sampling between the instruments**."

R13: Page 5. Line 14. ACSM data capture was 80% during deployment at Melpitz.

A13: Text was corrected as suggested

R14: Line 18. CDCE is Composition Dependent Collection Efficiency as used in the original Middlebrook et al paper. It is also important to note that CDCE algorithm does not take OM into account which can be a potential error source.

A14: We agree with the comments and the text was changed to: "The ACSM data was analyzed following the recommendation of manufacturer and applying **a composition dependent collection efficiency (CDCE) correction** based on the algorithms proposed by Middlebrook et al. (2012) to correct particle loss due to bouncing off the vaporizer before flash vaporization. **It is important to note that the CDCE algorithm includes inorganic species only and did not consider a possible effect of the organics on the collection efficiency estimation**".

R15: Line 23. … collect particles with size selective PM2.5 and PM10 inlets on preheated...

A15: replaced "cutting" by "**selective**"

R16: Line 24. Samples were collected on a daily basis...

A16: replaced "performed in" by "collected on"

R17: Page 6. Line 15. It is probably meant by "within Planetary Boundary Layer", but I do not quite understand what is meant by "above 500m" and why.

A17: The air mass trajectory analysis was set with an altitude of 500 m above the research station. Because Planetary Boundary Layer (PBL) is dynamic and its altitude is changing over time, it is critical to consider only air mass trajectories results that are connected with the measurements. In our case, if the PBL is located at an altitude below 500 m, the calculated air mass trajectory is reaching the station at an altitude above the PBL level. Consequently, this trajectory cannot be associated with the ground level measurements made at this time and will be not considered further in the analysis.

The text was changed as follows:

**"Finally, the meteorological conditions as available from the HYSPLIT output for each trajectory calculation were also examined. Although backward trajectories were started at a height of 500 m, the Planetary Boundary Layer at the trajectory starting time could be at a lower altitude making the association between the ground-based measurements and the inflowing air mass difficult. Therefore, only air mass trajectories with a HYSPLIT-estimated PBL height above 500 m were further considered for analysis."**

R18: Page 7. Line 2. ...mass concentration comparison is suggesting overestimation by ACSM when compared to offline PM1.

A18: The sentence was changed as suggested

"Over the entire period, the regression slope of the sulfate mass concentration comparison **is suggesting overestimation** by ACSM when compared to offline $PM_1$."

R19: Line 6. influenced by three specific periods in January 2015 and February 2017 (...).

A19: The sentence was corrected as suggested

R20: Line 10. The overestimation can be caused by either size selective inlet or(and) sodium sulfate mainly residing in coarse particles. However, I am not sure that is good enough explanation of very contrasting comparison between PM1 and PM2.5. Judging from slopes (1.45 for PM1 and 0.68 for PM2.5) that is suggesting 100% difference between PM1 and PM2.5 - that is massive and not necessarily related to ACSM.

A20: It is true that the overestimation of the ACSM sulfate mass concentration, when compared to $PM_{2.5}$ offline samples, can be caused by either the difference of the respective size inlet and/or presence of coarse sodium sulfate as mentioned on the manuscript ("This overestimation could be associated with the size-cutting difference between the two methods and the presence of not detected sulfate species on the coarse mode, such as sodium sulfate."). However, the difference between $PM_1$ and $PM_{2.5}$ is not as massive as suggested by the referee. On average over the 5.5 years, the ratio sulfate $PM_1$ to $PM_{2.5}$ is of 0.77, as can be seen in Figure SI-3. As it is discussed in the manuscript, the difference of slopes is strongly related to specific days showing an important coarse fraction as demonstrated by the sensitivity analysis (Fig. 3). By considering only days with a $PM_1:PM_{2.5} > 0.6$, the slopes between $ACSM-SO_4$ and $PM_1-SO_4$ is 0.96. consequently, the difference between the slopes for $PM_1$ and $PM_{2.5}$ agrees with the difference between the sulfate mass concentration on $PM_1$ and $PM_{2.5}$ samples.

R21: Line 18. That is correct theoretical explanation, but given huge discrepancy during "outlier" days the organic fragment signals should be of very significant magnitude to inorganic ones. Do authors have hints from the multiple campaigns when high resolution AMS was deployed at the site?

A21: We agree with the comment that the contribution of organic fragments to the sulfate signal is an absolute theoretical explanation and cannot be solved due to the unit mass resolution of the ACSM. However, based on the previous HR-ToF-AMS, we can confirm the presence of organic fragments on m/z 80 and 81. Based on the summer, fall, and winter HR-ToF-AMS measurements made at Melpitz and published in Poulain et al. (2011), $C_6H_9^+$, $C_5H_5O^+$ fragments were found at m/z 81 with concentrations closed to the one of $HSO_3^+$ on some time. The comparison between $UMR-SO_4$ and $HR-SO_4$ however, shows a different < 10 % for the campaigns published in Poulain et al., 2010.

As a result, the following sentence was added on the manuscript:

"Therefore, an increase of the organic signal at this m/z might lead to an overestimation of the ACSM sulfate mass concentration. **Although our previous measurements using High-Resolution Time-of-Flight Aerosol Mass Spectrometer (HR-ToF-AMS) at Melpitz (Poulain et al., 2011) support the presence of organic fragments on the UMR sulfate signal. The difference between the sulfate mass concentration based on UMR (as for ACSM) and the one obtained on the high-resolution (i.e. excluding the contribution of organic fragments on the sulfate signal) is below 10% indicating a minor impact of the organic to the sulfate signal.**"

R22: Line 27. coincides, not corresponds.

A22: replaced

R23: Line 31. PM1 cannot contribute to PM2.5 it is an inherent part of it. Is it not simply a ratio?

A23: Yes, it is. We changed the sentence by replacing "$PM_1$ contribution to $PM_{2.5}$" by "**$PM_1:PM_{2.5}$ ratio**" and homogenize the nomenclature on the entire manuscript.

R24: Page 8. Line 1. Consider that wet PM1 size segregation will inevitably pass less submicron particles. However, if that is the plausible culprit then ACSM should perfectly compare with PM2.5 which even in wet cut-off conditions should well correspond to dry ACSM PM1.

A24: This problem is inherent to all aerosol inlet system. However as discussed before on our answer A1, the cut-off shift due to aerosol hygroscopicity growth should play a minor role at Melpitz

The following sentence has been added to the text: "Therefore, the discrepancy between the ACSM and the $PM_1$ can be attributed to the individual upper size cutting of the two instruments, and it highlights the limits of such a comparison. **As already mentioned, a minor effect of the RH to the cut-off shift of the offline samplers can be expected at Melpitz (Chen et al., 2018)."**

R25: Line 5. The effect may still be there, but masked by the evaporation effect and NaNO3 effect on filters. Is this sentence out of place as it is becomes confusing whether the following sentences regard to sulfate or nitrate?

A25: We agree with the comment and the corresponding sentence was removed.

R26: Line 19. Please explain what air mass density means.

R26: The air mass density is a proxy of the occurrence of the trajectory within each cell. For clarity, it was replaced by the term "**trajectory density**", as defined in Petit et al. (2017).

R27: Line 20. This confirms the predominantly shallow PBL...

A27: Corrected

R28: Line 28. ...to slightly overestimate... The overestimation is very small and within the error margin compared to massive overestimation in Minguillon et al. I see this as a contrast from previous studies, not similarity.

A28: The text has been changed as follows: "The ACSM nitrate mass concentration tends to **slightly** overestimate the offline PM$_1$ nitrate throughout the entire period (slope = 1.16, R² = 0.80; Fig. 1 and 2). **This overestimation is very small and within the error margin compared to massive overestimation in** Ripoll et al. (2015) with a slope of 1.35 (R² = 0.77) and Minguillon et al. (2015) with a slope 2.8 (R² = 0.80)."

R29: Page 9. Line 5. I see more evidence in Figure 5. First, a lot of nitrate evaporation is visible at low ambient temperature as many of the points are clearly above ratio of 1.0. That is not surprising considering evaporation from particles already collected on the filter. However, the effect in summer seems to be larger, but concentrations are much lower in summer suggesting that evaporation can be near constant (as long as filter temperature is maintained similar during all seasons which needs to be confirmed). As more nitrate loaded particles are deposited on filter during winter they become buried under new layers of particles before being significantly evaporated. During summer there is little nitrate on particles and less particles altogether resulting in more absolute evaporative losses until particles buried under the new layers of particles.

A29: We thank the referee for his/her pertinent comments. The temperature effect described in figure 5 is linked to the evaporation from particles already collected on the filter only. The high-volume samplers are sitting outside and are not temperature controlled. Consequently, the inside temperature of the sampler is depending as well as of the outside temperature but also of the solar radiation that can contribute to warm-up the instrument. Both factors might explain the fact that the discrepancy between the two methods is already visible at low ambient temperature. Additionally, this leads to different inside temperature conditions during winter and summer periods making impossible a direct comparison of the evaporation process between summer and winter. This is the reason why we plotted Figure 5 using outside temperature.

 The following text has been added on the manuscript:

"In an intercomparison study of different sampling supports, Schaap et al. (2004) demonstrated that a quartz filter (PM$_{2.5}$ and PM$_{10}$) is a suitable material for sampling nitrate as long as temperature does not exceed 20 °C. **The high-volume samplers are sitting outside and are not temperature controlled. Therefore, the inside temperature of the sampler is influenced by the outside temperature.** This temperature artefact is clearly illustrated in Figure 5, when the variation of the ACSM:PM$_1$ nitrate ratio and the maximum temperature measured during the sampling day are compared. For ambient maximum temperatures above 10 °C, an increase of the ACSM:PM$_1$ ratio can be observed. Here it is imperative to note that the ambient maximum temperature did not reflect the temperature inside the sampler, **solar radiation may also contribute to warm up the sampler**."

R30: Page 10. Line 20. ...with off-line measurements fall somewhere in between the two previously discussed ions.

A30: corrected as suggested

R31: Page 12. Line 10. in another location.

A31: replaced "on" by "in"

R32: Line 25. The first comparison is correct by comparing mass with mass, but the second comparison is not, because particle mass derived from PNSD and MPSS will rely on ACSM for estimating average density making the two variables dependent. Therefore, only volume, not mass comparison of ACSM-MAAP and PNSD can be done correctly.

A32: Please see our previous answer (A2) on this comment for more details on this topic.

The sentence has been reworded as follows:

"The resulting total $PM_1$ mass, later referred to as the ACSM-MAAP-derived mass concentration, was then compared to the particle mass concentration obtained by weighting filters ($PM_1$ and $PM_{2.5}$) as well as to the calculated particle **volume and** mass concentration from the PNSD of the MPSS.

R33: Page 13. Line 3. Therefore, it is not possible to conclude whether....depends on the location or the presence of larger coarse mode.

A33: the sentence was corrected as suggested.

The text has been changed as follows: "**Therefore**, it is not possible to conclude whether this difference in correlation results between the two studies **depends on the** location or the presence of more coarse mode. Moreover, a possible loss of the more volatile compounds."

R34: Line 9. … cannot typically account for the entire mass

A34: The sentence was corrected as suggested.

R35: Line 12. It is inappropriate to arbitrarily choose specific OM:OC ratio when the above paragraphs discussed wildly different ratios.

A35: We agree that it would make more sense to apply a time-depend OM:OC ratio to convert the OC into OM. However, applying a time-depend OM:OC ratio based on the results discussed in the manuscript could lead to the introduction of bias on the offline mass closure. Indeed, it could be seen as directly replacing the OC results by the ACSM organic mass concentration. However, and as discussed in section 3.1.4, the ratio ACSM organic to offline OC is not free of artefact and uncertainties. The more key ones are the RIE-CE and Pieber effect for the ACSM, and the evaporation of volatile organic compounds during warm days for the filter samples. Moreover, the total PM mass concentration measured on the filter samples also include this temperature effect. Therefore, replacing the OC by the ACSM organic appears to be inappropriate for us. Nevertheless, and as mentioned by the referee, a constant OM:OC ratio could be considered as a too simple approach regarding the variability of the ratio. In consequence and to consider the aforementioned limitations, the seasonal OM:OC ratio presented on Figure 6-b was used since these values are resulting from a long dataset and should be less influenced by the day-to-day uncertainties.

Accordingly, the Figures 8 and SI-9 were replotted and the text changed as follows:" Here, the residual mass fraction was calculated as the difference between the weighted filter mass and the sum of the detected compounds **(Fig. SI-9). It is important to note here, that to properly convert the OC into OM and to consider all the different limitations inherent to both online and offline approaches, the seasonal means OM:OC ratio values (Fig. 6b) were applied**".

Figure caption SI-7 (now Fig. SI-9) was changed as follows: **"Time series of the Digitel PM$_1$ chemical composition. Conversion of OC into OM was made based on the seasonal OM:OC ratio presented in figure 6."**

R36: Line 32. Incorrect method. See above.

A36: Please refer to our answer A2 and the corresponding corrections/changes made on the manuscript.

R37: Page 14. Line 24. This artefact may plausibly explain the seasonality of the mass closure (it should be volume closure instead).

A37: The sentence was corrected as suggested and we refer to your previous answer (A2) regarding the discussion between volume and mass closure analysis

R38: Page 15. Line 1. By the same principle ACSM size range is 1um/1.6=0.625um (dvac/density=dmob) and discrepancies between size ranges of ACSM and MPSS should be irrelevant. Please reconsider and reflect in conclusions.

A38: The vacuum aerodynamic diameter of 1 μm indeed corresponds to approximately 625 nm (assuming a constant density of 1.6, spherical particle under free molecular regime). However, this assumption considered several approximations making the comparison potentially inaccurate. First, it assumes that the size cutting of the ACSM is absolute at 1 μm, which is not the case. The transmission of the aerodynamic lenses at 1 μm is ranging from 40 to 60% depending on the lens and the pressure (Liu et al., 2007). Moreover, and as already mentioned by the referee, the density is not constant over time, resulting in a time dependence of the equivalent diameter, which is ranging for Melpitz after monthly averaging over the entire period, from ca. 600 and 700 nm. Consequently, following the suggestion, it would be necessary to consider a time dependence on the upper size range for the MPSS to be properly accurate. Last but not least, applying such an approach will bring us back to the discussion on the volume- / mass-closure analysis bringing a dependency on the ACSM measurements to the MPSS dataset. This dependency might strongly impact the results since it is considered twice one for rescaling the MPSS measurement and second for the unit conversion (ACSM to volume or MPSS to mass).

Consequently, applying such correction on the MPSS data would certainly improve the correlation between the two instruments by smoothing the respective size effect, but it might be better for the wrong reasons. Then again, for certain specific environments, like a station with high coarse mode concentration (marine, dust environment), assumptions for cutting the MPSS scan-range before comparing with ACSM would make sense to avoid interference from super-micrometer multiple charged particles. This is important in case there is not APSS measurement available to perform a proper multiple charge correction.

R39: Line 31. Can this be called an artefact? It depends whether organo-nitrate belongs to organics class or nitrate. Nitrate functional group is still a nitrate even if bound to organic species. Quite contrary, off-line inorganic and offline OC/EC analysis completely misses nitrate bound to organics making the use of the term "artefact" justified. Ideally, one would want a clear distinction of organo-nitrate compound which would be out of reach by off-line inorganic techniques unless specifically measured for OrgNO3.

A39: We agree with the comment and replace "artefact" by **"uncertainty"**.

R40: Figure 1. I recommend changing PM to ACSM-MAAP or ACSM+eBC for consistency with further Figures.

A40: We disagree with the suggestion as Figure 1 shows the time series of the particulate mass concentration from the ACSM-MAAP but also PM mass concentration of the filter samples. Therefore, the use of PM appears to us as the most relevant.

R41: Figure 2. ...and solid black lines represent regression fit by least orthogonal distance (y=a+bx)

A41: corrected

R42: Figure 3. change "data coverage" to "data capture".

A42: We updated Figure 3

R43: Figure 4. ...for days where sulfate concentration difference between PM2.5 and PM1 exceeded 1ug/m3: (a) overpassing trajectory density; (b) potential source contribution function??; (c) time series of sulfate concentration difference, PBL height above 2000m, precipitation events exceeding 1mm/h and PBL formation above the station altitude.

A43: Text was changed as follows: "**Trajectory analysis for days where sulfate concentration difference between PM$_{2.5}$ and PM$_1$ exceeded 1 μg m$^{-3}$: (a) overpassing trajectory density; (b) results of the potential source contribution function (PSCF) analysis; (c) time series of sulfate mass concentration difference, trajectory altitude above 2000 m, precipitation events exceeding 1 mm h$^{-1}$ and PBL above the station < 500 m.**"

R44: Figure 6. ...and off-line OC PM1 (a); seasonal variability....(b) and the entire time series (c).

A44: Corrected

R45: Figure 7. Awkward Figure caption. Please rewrite according to suggested above.

A45: Figure caption was rewritten as follows:

"**Correlations between the estimated OC$_{ACSM}$ and the offline OC mass concentration over the entire period and seasonality for PM$_1$ (a), and PM$_{2.5}$ (b). Black lines show the least orthogonal linear fit and the red dotted lines the 1:1 line.**"

R46: Figure 9. Incorrect graph as it should be volume comparison. Awkward Figure caption. Please rewrite according to suggested above.

A46: The figure was modified as follows:

"**Comparison between measured ACSM-MAAP and MPSS for the entire period and seasonal variability: volume-closure (a), mass-closure (b), median number size distribution (red) with 10-90 (grey line) and 25-75 (black boxes) percentiles (c), median volume size distribution (d). The linear regressions (red lines) were calculated using the least orthogonal distance fit method.**".

R47: Figure S1. Make PM axis consistent with further Figures

A47: Please refer to our answer A40 which is dealing with the same comment

R48: Figure S2. ...to daily PM2.5 mass.

A48: Figure caption was changed to "**Scatter plot of the ACSM species mass concentration measurements compared to corresponding daily PM$_{2.5}$ mass concentration over the entire period and seasonality.**"

R49: Figure S5. ...corrected for organo-nitrate contribution during days with PM1:PM2.5 < 0.6

A49: The figure caption was changed to "**Scatter plot of the winter (DJF) ACSM nitrate mass concentration corrected for organo-nitrate contribution during days with a PM$_1$:PM$_{2.5}$ < 0.6 compared to PM$_1$ nitrate mass concentration. The linear regressions (red lines) were calculated using the least orthogonal distance fit method.**"

**Abstract :**

Line10 : replaced : « The Aerosol Chemical Speciation Monitor (ACSM) is an instrument for identifying and quantifying the influence of air quality mitigations. For this purpose, a European ACSM network has been developed within the research infrastructure project ACTRIS (European Research Infrastructure for the observation of Aerosol, Clouds and Trace Gases). To ensure the uniformity of the dataset, as […]"
By "The Aerosol Chemical Speciation Monitor (ACSM) is nowadays widely used to identify and quantify the main component of fine particles in ambient air. As such, its deployment at observatory platforms is fully incorporated within the European Aerosol, Clouds and Trace Gases Research Infrastructure (ACTRIS). To ensure the consistency of the dataset […]"

Line 16: added the following sentence: "Here, we present and discuss the main outputs of long-term quality assurance efforts achieved for ACSM measurements at the research station Melpitz (Germany) since 2012 onwards."

Line 16: change the sentence "In order to check the robustness of the ACSM over the years and to characterize the seasonality effect, nitrate, sulfate, ammonium […]" by "In order to validate the ACSM measurements over the years and to characterize seasonal variations, nitrate, sulfate, ammonium […]"

Line 22: added "Mobility Particle size Spectrometer (MPSS)"

**1. Introduction**

Page 2, Line 22: modified and added a sentence "For this purpose, a European network of ground-based Aerosol Chemical Species Monitor (ACSM, Ng et al., 2011) is operated within the European Research Infrastructure ACTRIS (European Research Infrastructure for the observation of Aerosol, Clouds and Trace Gases, http://www.actris.eu). One of the main objectives of this coordinated ACSM network is to investigate and […]"
by ". For this purpose, a European distributed facility of ground-based Aerosol Chemical Species Monitor (ACSM, Ng et al., 2011) is operated within ACTRIS (European Research Infrastructure for the observation of Aerosol, Clouds and Trace Gases, http://www.actris.eu). Complementary, the COST Action CA16109 Chemical On-Line cOmpoSition and Source Apportionment of fine *aerosol* (COLOSSAL, https://www.costcolossal.eu) is gathering a wide community of European research groups (with even further international inputs, as well as participation of some regional air quality monitoring networks) interested in the fine aerosol fraction. One of the main objectives of these coordinated programs is to investigate and […]"

Page 3, Line 8: replace "Ovadnevaite et al. (2014) have written a rare published work that reports long-term AMS comparisons." By "Ovadnevaite et al. (2014) have written a rare published work that reports long-term AMS comparisons (3-years). Even though the authors successfully compared it with offline $PM_{2.5}$ filter samples, they did not discuss the variability of the correlation and potential source of uncertainties.

Page 3, Line 18: replace "focus" by "Scope"

**2.1 Research observatory Melpitz**

Page 4, line 9: replace "effect" by "impact"
Line18 added "actively" in "[…] automatic aerosol diffusion dryer to actively keep the relative humidity […]"

Page 4, Line 18: Replaced the sentence "The aerosol flow is divided among a set of instruments" by "The aerosol flow is divided among a set of instruments by an isokinetic splitter (WMO/GAW, 2016) ensuring a representative sampling between the instruments".

**2.2 ACSM**

Page 5, Line 14: replace "covers" by "capture"

Page 5, Line 15: the following text has been added "The ACSM was regularly calibrated according to the manufacturer's recommendations at that time with 350 nm monodispersed ammonium nitrate and ammonium sulfate particles selected by a DMA and using the jump scan approach. It is important to note that since recently, the recommended calibration method has changed to a full scan approach (Freney et al., 2019). The total particle number concentration was systematically set below $800 \, \# \, cm^{-3}$ to limit the artefact due to multiple charged particles. An overview of the ionization efficiency (IE) and relative ionization efficiency (RIE) for ammonium and sulfate can be found in Figure SI-1. On average, all performed calibration provides a mean IE value of 4.93 ($\pm$1.45) $10^{-11}$ (mean $\pm$ std. dev.) and mean RIEs for ammonium and sulfate were 6.48$\pm$1.26 and 0.68$\pm$0.13, respectively. These values are very close to the ones used for the data evaluation as indicated in Figure SI-1. Overall, no clear trend for IE and RIE of sulfate can be observed over the period, while a small decrease in the RIE of ammonium can be reported. The lowest RIE of ammonium was reported just after the replacement of the filament indicated a possible need for degassing and stabilization period. However, it is difficult to conclude if these tendencies could be associated with a possible aging effect of the instrument since it corresponds to a single instrument. Similar observations on various other individual ACSMs would be needed to allow for stating such a conclusion and a more systematic investigation of potential trends should then be performed with a large number of ACSM."

Page 5, Line 18: replaced "chemical time-dependent collection efficiency" by "composition dependent collection efficiency (CDCE)"

Page 5, Line 19: the following sentence has been added ". It is important to note that the CDCE algorithm includes inorganic species only and did not consider a possible effect of the organics on the collection efficiency estimation."

**2.4 Air mass trajectory analysis**

Page 6, line 14: The sentence "Finally, the meteorological conditions as available in HYSPLIT during each trajectory were also examined and only the trajectories ending with a Plenary Boundary Layer height (PBL) above 500 m were further considered for analysis" was replaced by "Finally, the meteorological conditions as available from the HYSPLIT output for each trajectory calculation were also examined. Although backward trajectories were started at a height of 500 m, the Planetary Boundary Layer at the trajectory starting time could be at a lower altitude making the association between the ground-based measurements and the inflowing air mass difficult. Therefore, only air mass trajectories with a HYSPLIT-estimated PBL height above 500 m were further considered for analysis."

**3.1 Comparison with offline chemical composition**

Page 6, line 29: The following sentence was added: "It is also important to note here that the comparison between ACSM and offline samplers generally consists of comparing dry aerosol online measurements to offline analyses of samples collected at ambient RH. A direct consequence is that the offline results might suffer from a cut-off shift due to aerosol hygroscopic growth when ambient RH is high (Chen et al., 2018). Based on this study, the cut-off shift due to aerosol hygroscopic growth should play a minor role at Melpitz, as this effect was estimated to influence the comparison by 2 % for marine air-mass and 1 % for continental air-mass. For European background stations, such a cut-off shift has been estimated to represent less than 10 % for $PM_1$ and 20 % for $PM_{2.5}$ particle

mass loading, while it is stronger for marine or coastal stations (up to 43 % for $PM_1$ and 62 % for $PM_{2.5}$). Therefore, such artefact has to be considered when comparison ACSM with offline measurements."

**3.1.1 Sulfate**

Page 7, line 19: The following sentence was added ". Although our previous measurements using High-Resolution Time-of-Flight Aerosol Mass Spectrometer (HR-ToF-AMS) at Melpitz (Poulain et al., 2011) support the presence of organic fragment on the UMR sulfate signal. The difference between the sulfate mass concentration based on UMR (as for ACSM) and the one obtained on the high-resolution (i.e. excluding the contribution of organic fragments on the sulfate signal) is below 10% indicating a minor impact of the organic to the sulfate signal."

Page 8, line 3: the following sentence was added "As already mentioned, a minor effect of the RH to the cut-off shift of the offline samplers can be expected at Melpitz (Chen et al., 2018)."

Page 8, line 5: The following sentence was removed: ". Interestingly, the PM1:PM2.5 ratio has a minor influence on nitrate and OC correlation parameters, as will be discussed later on".

**3.1.2 Nitrate**

Page 8, line 29: Replaced "Such an overestimation of nitrate mass concentrations by the ACSM has already been shown by Ripoll et al. (2015) […]" by "This overestimation is very small and within the error margin compared to massive overestimation in […]"

Page 9, line 4: Added "The high-volume samplers are sitting outside and are not temperature controlled. Therefore, the inside temperature of the sampler is influenced by the outside temperature."

**3.1.3 Ammonium**

[revised manuscript text omitted]

**3.2.1 Mass closure with offline filters**

Page 13, line 10: the sentence "Here, the residual mass fraction was calculated as the difference between the weighted filter mass and the sum of the detected compounds applying a constant OM:OC ratio of 1.8 to convert OC into OM (Fig. SI-7)" was rewritten as follows "Here, the residual mass fraction was calculated as the difference between the weighted filter mass and the sum of the detected compounds (Fig. SI-9). It is important to note here, that to properly convert the OC into OM and to consider all the different limitations inherent to both online and offline approaches, the seasonal means OM:OC ratio values (Fig. 6b) were applied.".

**3.2.2 Mass closure with PNSD**

Page 13: the section from page 13 line 29 to page 14 line 12 was rewritten as follows: "The PNSD has been continuously measured in parallel to the aerosol mass spectrometer and can, therefore, be used to perform mass closure analysis between ACSM-MAAP and PNSD (ranging from 10 to 800 nm, mobility diameter). To ensure a robust comparison between the two systems, two approaches are reported in the literature: the first one consists of converting the ACSM-MAAP mass concentration into volume and the PNSD in volume concentration. The second one consists of converting the PNSD into mass concentration. Both approaches are based on the same assumptions of (i) spherical, (ii) fully internally mixed particles, and (iii) an identical chemical composition over the entire size distribution to estimate a chemical time-dependent gravimetric particle density based on the following equation from Salcedo et al. (2006):

$$density = \frac{[Total_{AMS} + eBC]}{\frac{[NO_3^-] + [SO_4^{2-}] + [NH_4^+]}{1.75} + \frac{[Cl^-]}{1.52} + \frac{[Org]}{1.2} + \frac{[eBC]}{1.77}} \qquad (1)$$

Here, the density was assumed to be 1.75 g cm$^{-3}$ for ammonium nitrate and ammonium sulfate (Lide, 1991), 1.52 g cm$^{-3}$ for ammonium chloride (Lide, 1991), and 1.2 g cm$^{-3}$ for organic matter (Turpin and Lim, 2001). Finally, a density of 1.77 g cm$^{-3}$ (Park et al., 2004) was applied for eBC. A discussion of eBC density can be found in Poulain et al. (2014).

It is important to note that for the volume concentration approach, both measurements (ACSM-MAAP and MPSS) remain independent between each other, which is not the case when using mass concentration. However, only a few numbers of papers reported a comparison between AMS or ACSM and MPSS in volume concentration (e.g. DeCarlo et al., 2008;Elsasser et al., 2012). Even though the two variables are non-independent on the mass concentration approach, it remains the most commonly used. A possible reason is that the mass concentration unit remains easier to use and interpreted as the volume concentration since atmospheric measurements are usually made in mass concentration.

Here, we investigated comparison results obtained using each of these approaches. Results are summarized in Figure 9.a for the volume concentration approach and Figure 9.b for the mass concentration one. Over 5.5 years of measurements, the ACSM-MAAP-derived volume or mass concentration correlates well with the estimated volume or mass concentration of the MPSS with the similar slopes of 0.79 and 0.77, respectively (R² = 0.90, Fig. 9a and 9b). This matches similar previous comparisons at the same place with an AMS (Poulain et al., 2014). Therefore, the selected method (volume or mass) did not substantially influence the comparison results. In the conditions of the present study both approaches could be applied for the station of Melpitz. Since comparison in mass concentration is the more commonly used, we will focus on it in the following discussions."

**4   Summary and conclusion**

Page 15, line 16: Added: "For such an exercise, it is fundamental to ensure isokinetic flow splitting between the different instruments connected to the main sampling line to ensure a homogeneous distribution of the air sample."

Page 16, line 9: Replaced "Nevertheless, more systematic comparisons should be performed in a similar way in different environments to validate our results and to better identify f44 instrumental variability" by "Nevertheless, the method might be difficult to apply for short time measurements (e.g. a few weeks only), where low/high extreme ratios may be misinterpreted, and results interpreted with cautions, such OA-OC comparison and OC$_{ACSM}$:OC$_{offline}$ methods shall then preferably be used on long term continuous measurements. Finally, it should also be noted that OC is the only regulated organic aerosol-related variable commonly monitored within current air quality networks (Directive 2008/50/CE, 2008;WMO/GAW, 2016) whereas equivalent methods for a better OA quantification at high-time resolution are still to be standardized, reinforcing the need for much more systematic comparison exercises at various locations."

Page 16, line 15: Added the following sentence "Moreover, possible cut-off shift due to ambient relative humidity effect on the offline measurements could represent a non-negligible parameter and has to be considered during such an exercise, especially for marine stations."

Page 16, line 22: Added the following sentence "It can be performed by converting the ACSM-MAAP mass concentration into volume concentration or by converted the MPSS volume concentration into mass concentration both using time-dependent density and assuming spherical and fully internally mixed particles. The volume approach is the most robust since it enables a strictly independent method. Being more interpretable, the mass approach may be used instead of for error quantification as long as it agrees with the volume approach. For the present dataset, the selected method did not substantially influence the comparison results."

**Figures**

Figure 3: corrected labelling right y-axis

Figure 4: replaced the figure capture by "Trajectory analysis for days where sulfate concentration difference between $PM_{2.5}$ and $PM_1$ exceeded 1 µg m$^{-3}$: (a) overpassing trajectory density; (b) results of the potential source contribution function (PSCF) analysis; (c) time series of sulfate mass concentration difference, trajectory altitude above 2000 m, precipitation events exceeding 1 mm h$^{-1}$ and PBL above the station < 500 m."

Figure 6: the figure was corrected and the figure caption was changed to "Correlation between ACSM organic mass concentrations and offline OC $PM_1$ (a); seasonal variability of the estimated $OM_{ACSM}$:$OC_{PM1}$ ratio (b), and the entire time series colored by maximum daily temperature (c)."

Figure 7: the figure caption was changed to "Correlations between the estimated $OC_{ACSM}$ and the offline OC mass concentration over the entire period and seasonality for $PM_1$ (a), and $PM_{2.5}$ (b). Black lines show the least orthogonal linear fit and the red dotted lines the 1:1 line."

Figure 8 The figure was corrected

Figure 9 : the figure was remade and the figure caption was changed accordingly to "Comparison between measured ACSM-MAAP and MPSS for the entire period and seasonal variability: volume-closure (a), mass-closure (b), median number size distribution (red) with 10-90 (grey line) and 25-75 (black boxes) percentiles (c), median volume size distribution (d). The linear regressions (red lines) were calculated using the least orthogonal distance fit method. "

**Supplementary information**

Co-authors and corresponding affiliation lists were corrected.

Two new figures were added:

[Figure]

Figure SI-1: Time variation of the IE and RIE for ammonium and sulfate. The single points correspond to calibration, the dashed black line to the mean value from the calibration, and the full red line the mean value from the data analysis (shaded area corresponds to the standard deviation). Major maintenance (change of filament and vaporizer) are including.

[Figure]

Figure SI-7: Comparison of the measured ammonium with the predicted ammonium mass concentration for the ACSM (left) and offline PM$_1$ samples assuming a full neutralization by nitrate, sulfate, and chloride. The linear regressions were calculated using the least orthogonal distance fit method (y = a x + b).

Figure SI-2: the figure caption was changed as follows "Scatter plot of the ACSM species mass concentration measurements compared to corresponding daily PM$_{2.5}$ mass concentration over the entire period and seasonality."

Figure SI-5: the figure caption was changed as follows "Scatter plot of the winter (DJF) ACSM nitrate mass concentration corrected for organo-nitrate contribution during days with a PM$_1$:PM$_{2.5}$ < 0.6 compared to PM$_1$ nitrate mass concentration. The linear regressions (red lines) were calculated using the least orthogonal distance fit method."

Figure SI.7: the figure was corrected according to referee comment and the figure caption updated accordingly as follows "Time series of the Digitel PM$_1$ chemical composition. Conversion of OC into OM was made based on the seasonal OM:OC ratio presented in Figure 6."

[revised manuscript text omitted]

**Figure SI-1: Time variation of the IE and RIE for ammonium and sulfate. The single points correspond to calibration, the dashed black line to the mean value from the calibration, and the full red line the mean value from the data analysis (shaded area corresponds to the standard deviation). Major maintenance (change of filament and vaporizer) are including.**

**Table SI-1:** Fitting parameters of the comparison of the mass concentrations measured by the ACSM and offline $PM_1$ considering the identified threshold $PM_1:PM_{2.5} < 0.6$ or the entire $PM_1$ dataset.

| | | $PM_1$ (threshold $PM_1:PM_{2.5} < 0.6$) | | | $PM_1$ no threshold | | |
|---|---|---|---|---|---|---|---|
| species | Time period | Slope | Intercept | R² | Slope | Intercept | R² |
| Total mass | All dataset | 1.02 | -4.21 | 0.90 | 1.29 | -6.64 | 0.72 |
| | Winter | 1.01 | -3.27 | 0.87 | 1.3 | -5.44 | 0.77 |
| | Spring | 0.96 | -2.82 | 0.89 | 0.97 | -2.85 | 0.83 |
| | Summer | 0.89 | -3.59 | 0.40 | 0.9 | -3.54 | 0.35 |
| | Fall | 1.31 | -6.83 | 0.67 | 1.55 | -8.79 | 0.58 |
| | | | | | | | |
| Sulfate | All dataset | 0.96 | -0.06 | 0.77 | 1.45 | -0.63 | 0.59 |
| | Winter | 0.98 | -0.06 | 0.83 | 1.57 | -0.43 | 0.61 |
| | Spring | 1.01 | -0.14 | 0.72 | 0.98 | -0.08 | 0.74 |
| | Summer | 0.85 | 0.00 | 0.80 | 0.87 | -0.01 | 0.77 |
| | Fall | 1.06 | -0.06 | 0.54 | 1.25 | -0.29 | 0.58 |
| | | | | | | | |
| Nitrate | All dataset | 1.16 | 0.65 | 0.80 | 1.32 | 0.6 | 0.79 |
| | Winter | 1.29 | 0.04 | 0.80 | 1.49 | -0.06 | 0.74 |
| | Spring | 1.05 | 0.88 | 0.82 | 1.09 | 0.85 | 0.81 |
| | Summer | 6.28 | -0.29 | 0.29 | 6.47 | -0.35 | 0.27 |
| | Fall | 1.63 | 0.55 | 0.51 | 1.9 | 0.39 | 0.56 |
| | | | | | | | |
| Ammonium | All dataset | 1.14 | -0.05 | 0.77 | 1.40 | -0.21 | 0.71 |
| | Winter | 1.02 | 0.08 | 0.83 | 1.36 | -0.09 | 0.65 |
| | Spring | 1.15 | -0.02 | 0.79 | 1.15 | -0.01 | 0.80 |
| | Summer | 1.47 | -0.37 | 0.49 | 1.53 | -0.39 | 0.51 |
| | Fall | 1.74 | -0.31 | 0.44 | 2.01 | -0.49 | 0.46 |
| | | | | | | | |
| Organic ($OM_{ACSM}$) | All dataset | 1.71 | 0.11 | 0.68 | 1.85 | -0.1 | 0.74 |
| | Winter | 1.29 | 0.28 | 0.78 | 1.70 | -0.29 | 0.76 |
| | Spring | 1.84 | 0.02 | 0.81 | 1.89 | -0.1 | 0.79 |
| | Summer | 2.74 | -1.44 | 0.68 | 2.83 | -1.58 | 0.66 |
| | Fall | 2.49 | -1.07 | 0.69 | 2.41 | -1.06 | 0.67 |

[Figure]

**Figure SI-2**: Time series ACSM (daily average, black line) and daily 24 h PM$_{2.5}$ filter samples (colored bars) for the total mass, sulfate, nitrate, and ammonium. The particulate matter (PM) corresponds to the sum of ACSM species and eBC$_{PM1}$ for the on-line instrument and the filter mass for the off-line samples.

[Figure]

**Figure SI 23**: Scatter plot of the ACSM species mass concentration measurements compared to corresponding daily PM2.5 mass concentration over the entire period and seasonality.

Scatter plot of the ACSM mass concentration measurements compared to daily mass in PM2.5.

[Figure]

**Figure SI-4**: Seasonal variability of the $PM_1:PM_{2.5}$ ratios of total PM (a), nitrate (b), sulfate (c) and OC (d).

[Figure]

**Figure SI-5:** **Influence of coarse mode particles on particle number and volume size distribution in the range 10 to 5000 nm obtained by merging MPSS and APSS (assuming spherical particle and a constant density of 1.6). The top panels represent the daily median particle number (black full line) and volume (red dots) size distribution for selected days emphasized by the white box on the particle volume size distribution from February 2017. The dashed black lines on the plots indicate the upper size bin of the MPSS.**

[Figure]

**Figure SI-6**: Scatter plot of the winter (DJF) ACSM nitrate mass concentration corrected for organo-nitrate contribution during days with a $PM_1:PM_{2.5} < 0.6$ compared to $PM_1$ nitrate mass concentration. The linear regressions (red lines) were calculated using the least orthogonal distance fit method.

[Figure]

**Figure SI-7: Comparison of the measured ammonium with the predicted ammonium mass concentration for the ACSM (left) and offline $PM_1$ samples assuming a full neutralization by nitrate, sulfate, and chloride. The linear regressions were calculated using the least orthogonal distance fit method (y = a x + b).**

[Figure]

**Figure SI-68: influence of the maximum day temperature to the OC$_{ACSM}$:OC$_{PM1}$ ratio.**

[Figure]

[Figure]

**Figure SI-9:** **Time series of the Digitel PM₁ chemical composition.** Conversion of OC into OM was made based on the seasonal OM:OC ratio presented in Figure 6.